genetics

polymer–polymer phase separation, Hi-C maps, epigenetics, block copolymers, HP1, H3K9me2/3

**Author for correspondence:**
Prim B. Singh
e-mail: prim.singh@nu.edu.kz

# On the relations of phase separation and Hi-C maps to epigenetics

Prim B. Singh[1,2] and Andrew G. Newman[3]

[1]Nazarbayev University School of Medicine, 5/1 Kerei, Zhanibek Khandar Street, Nur-Sultan Z05K4F4, Kazakhstan
[2]Epigenetics Laboratory, Department of Natural Sciences, Novosibirsk State University, Pirogov Street 2, Novosibirsk 630090, Russian Federation
[3]Institute of Cell and Neurobiology, Charité—Universitätsmedizin Berlin, Corporate member of Freie Universität Berlin, Humboldt-Universität zu Berlin and Berlin Institute of Health, Berlin, Germany

 PBS, 0000-0002-9571-0974; AGN, 0000-0002-0222-9162

The relationship between compartmentalization of the genome and epigenetics is long and hoary. In 1928, Heitz defined heterochromatin as the largest differentiated chromatin compartment in eukaryotic nuclei. Müller's discovery of position-effect variegation in 1930 went on to show that heterochromatin is a cytologically visible state of heritable (epigenetic) gene repression. Current insights into compartmentalization have come from a high-throughput top-down approach where contact frequency (Hi-C) maps revealed the presence of compartmental domains that segregate the genome into heterochromatin and euchromatin. It has been argued that the compartmentalization seen in Hi-C maps is owing to the physiochemical process of phase separation. Oddly, the insights provided by these experimental and conceptual advances have remained largely silent on how Hi-C maps and phase separation relate to epigenetics. Addressing this issue directly in mammals, we have made use of a bottom-up approach starting with the hallmarks of constitutive heterochromatin, heterochromatin protein 1 (HP1) and its binding partner the H3K9me2/3 determinant of the histone code. They are key epigenetic regulators in eukaryotes. Both hallmarks are also found outside mammalian constitutive heterochromatin as constituents of larger (0.1–5 Mb) heterochromatin-*like* domains and smaller (less than 100 kb) complexes. The well-documented ability of HP1 proteins to function as bridges between H3K9me2/3-marked nucleosomes contributes to polymer–polymer phase separation that packages epigenetically heritable chromatin states during interphase. Contacts mediated by HP1 'bridging' are likely to have been detected in Hi-C maps, as evidenced by the B4 heterochromatic subcompartment that emerges from contacts

between large KRAB-ZNF heterochromatin-*like* domains. Further, mutational analyses have revealed a finer, innate, compartmentalization in Hi-C experiments that probably reflect contacts involving smaller domains/complexes. Proteins that bridge (modified) DNA and histones in nucleosomal fibres—where the HP1–H3K9me2/3 interaction represents the most evolutionarily conserved paradigm—could drive and generate the fundamental compartmentalization of the interphase nucleus. This has implications for the mechanism(s) that maintains cellular identity, be it a terminally differentiated fibroblast or a pluripotent embryonic stem cell.

## 1. Introduction

Cursory inspection of eukaryotic nuclei using a simple light microscope shows that the optical density of chromatin is not uniform. On this basis, Heitz [1] defined heterochromatin as the dense compartment that is opaque to transmitted light and stains deeply with simple dyes, while euchromatin was the other compartment that stained lightly and through which light passed readily. Beyond this strictly empirical definition, quantitative techniques have shown that DNA in mammalian interphase nuclei is indeed more densely packed in constitutive heterochromatin compared with euchromatin. There is two to sixfold higher density of DNA in heterochromatin as measured by fluorescence intensity of DNA-binding fluorophores [2,3], which can be confirmed by measuring nucleosome density using fluorescently tagged histones [3,4]. The increased nucleosome density reflects how the 11 nm 'beads-on-a-string' nucleosome fibre is packaged in heterochromatin, and recent work provides a pathway that might explain the increased density observed. In H3K9me3-marked heterochromatin, the preferred contact geometry of the nucleosome fibre is a two-start helical fibre with stacked alternating nucleosomes, making the closest neighbour the second-nearest nucleosome rather than next nearest nucleosome [5]. Super-resolution imaging has revealed another level of organization that is characterized by the assembly of irregularly folded 'clutches' of nucleosomes where the density of larger 'clutches' is greater in heterochromatin compared to euchromatin [6]. The molecular crowding observed in the heterochromatic environment [2,4] has been modelled and predicted to enhance the interaction between the 'clutches' owing to osmotic depletion attraction [7]. The calculated entropy-driven attraction is small, approximately $0.5k_BT$, but could favour the merging of 'clutches' to form 'dense domains' or 'globules' that have been detected by super-resolution imaging [8] and chromosome conformation capture [9].

Soon after Heitz's definition of heterochromatin, Müller [10] discovered the phenomenon of ever-sporting displacements in *Drosophila*, later called position-effect variegation (PEV). PEV continues to be an important experimental paradigm for interrogating the relationship of constitutive heterochromatin to euchromatin in a living animal by disrupting the natural boundary that separates the two cytologically distinguishable states of chromatin (reviewed in [11–14]). PEV led to key conceptual advances and generated invaluable molecular tools that have done much to provide an outline of the natural history of constitutive heterochromatin by unveiling conserved mechanisms that operate in species ranging from fission yeast to man [15,16]. Outstanding among the contributions of PEV were, first, the demonstration that the effect of constitutive heterochromatin on gene repression is pervasive and heritable. Pervasive because, in most cases of PEV, repression results from 'spreading' of the dense packaging from within constitutive heterochromatin across the variegating breakpoint into euchromatin [17–19]. Once established, repression is heritable from one cellular generation to the next [20,21]. Thus, the discovery of PEV [10] gave birth to the discipline of *epigenetics* more than a decade before the term itself was coined [22]. Second was the identification of second-site modifiers of variegation that encode structural and enzymatic components of constitutive heterochromatin (reviewed in [14,23]). Two of these modifiers encode proteins that are highly conserved in organisms from fission yeast to man. One is heterochromatin protein 1 (HP1) and the other H3K9 HMTases that generate the H3K9me2/3 determinant of the histone code to which HP1 binds [24]. HP1 and H3K9me2/3 are hallmarks of constitutive heterochromatin and key epigenetic regulators in eukaryotes [15,16]. They represent a potential link between compartmentalization and epigenetics that will be explored in this paper. We now turn to these hallmarks with a focus on mammalian HP1 proteins because recent *in vitro* work has indicated that they form liquid–liquid condensates and gel-like states [25–27] that could drive compartmentalization of cytologically visible constitutive heterochromatin in interphase nuclei.

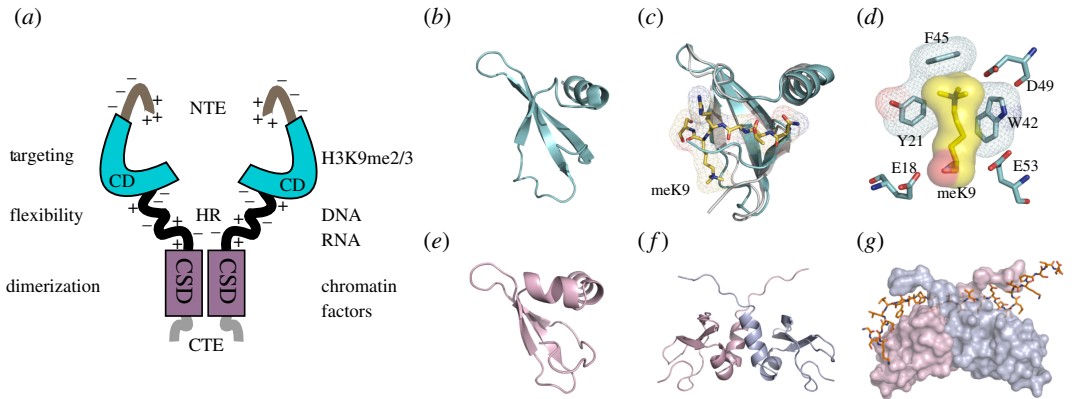

**Figure 1.** Structure–function relationships of mammalian HP1β. Depicted are the functional properties of the HP1β domains, its interaction with H3K9me3 and the PxVxL penta-peptide motif. (*a*) Cartoon summarizing the functional properties (left) and interactions of different domains of hHP1β (right). The specificity of HP1β CD interaction with H3K9me3-marked chromatin can be modulated by the negatively and positively charged residues within the NTE and HR [31]. (*b*) The HP1β CD (PDB code 1GUW) forms a three-stranded β-pleated sheet that abuts to a α-helix. The site of the shallow groove to which the H3-tail binds is between the third strand, a C-terminally adjacent coil segment and the N-terminal segment (note: this conformation corresponds to the peptide-bound complex). (*c*) The HP1β CD (1GUW; cyan) complexed with the H3K9me3 tail peptide (1KNA; yellow stick) superimposed on the HP1β CD alone (1AP0; grey). Binding of the H3K9me3 tail peptide causes the CD N-terminal region to draw upwards and wrap around the peptide. meK9, methyl-ammonium group. (*d*) A consequence of this induced fit is that a notional aromatic 'cage' is formed from three conserved aromatic residues: Tyr21, Trp42 and Phe45. The interaction between the methyl-ammonium moiety and the aromatic cage is largely electrostatic and mediated by cation–π interactions where the positively charged (cation) moiety is attracted to the negative electrostatic potential of the aromatic groups' π-system [32]. (*e*) The HP1β CSD monomeric subunit (PDB code 1SZ4) [33] shows a similar mixed α/β fold as the HP1β CD, except for an additional α-helix that is shown facing the reader. The CSD has a groove corresponding to that found in the CD, but is partly occluded (in the vicinity of the putative K9 binding site) by the N-terminal residues of the CSD. (*f*) Structure of the HP1β CSD dimer (1SZ4). The monomers have an affinity of $K_D$ approximately 150 nM; homodimer formation mainly involves interactions between the α2 helices. The dimer creates a groove between the first β-strand and the C-terminal segment at the CSD–CSD interface which can bind proteins that possess the PxVxL motif [33,34]. (*g*) Surface view of the CSD homodimer (one monomer in pink and the other in blue) bound to the CAF-1 peptide (shown as a stick model) containing the PxVxL motif, which is involved in intermolecular β pairing with both monomers [33]. (*a*) was modified from [31]. (*b*)–(*g*), with legend, were taken from [35].

## 2. Mammalian heterochromatin protein 1 proteins and polymer–polymer phase separation

In mammals, there are three HP1 isotypes, termed HP1α, HP1β and HP1γ, which are encoded by distinct genes, chromobox homologue 5 (*Cbx5*), *Cbx1* and *Cbx3*, respectively [28]. Immuno-localization studies have shown that HP1α and HP1β are usually enriched within constitutive heterochromatin [29], where their concentration is around 10 μM [3]. HP1γ has a more euchromatic distribution [29]. They are small approximately 25 kDa molecules that consist of two globular domains, an N-terminal chromo domain (CD) and a sequence-related C-terminal chromo shadow domain (CSD), linked by an unstructured, flexible, hinge region (HR) [30]. Depending on the species and isoforms, there are less well-conserved N- and C-terminal extensions (NTE and CTE, respectively) (figure 1*a*). The CD specifically binds to the N-terminal tail of histone H3, when methylated at the lysine 9 residue (H3K9me) [36–38], with the highest (μM) affinity for the *tri*-methylated form (H3K9me3; [39]; figure 1*b*,*c*). From the crystallographic data, the methyl-ammonium group in K9H3 is caged by three aromatic side chains in the CD, where the binding energy is driven largely by cation–π interactions [32] (figure 1*d*). The CSD dimerizes and forms a 'nonpolar' pit that can accommodate penta-peptides with the consensus sequence motif PxVxL, found in many HP1-interacting proteins [33,34,40] (figure 1*e*–*g*). There are likely to be other modes of interaction with the nucleosome including, for example, that of the HP1 CD or CSD with the H3 histone 'core' [41–43], binding of the HR region to DNA and RNA [44–46] and a non-specific electrostatic interaction of the NTE with the H3 tail [31].

Mutational analysis in mice has shown that mammalian HP1 isotypes have different mutant phenotypes despite sharing extensive sequence identity [28,47]. HP1α function is essentially redundant. $Cbx5^{-/-}$ mice are viable and fertile (cited in [48]) [47] albeit they exhibit a very specific defect where $T_H1$-specific gene expression is not silenced in $T_H2$ cells [49]. HP1β function cannot be compensated by HP1α and γ. The $Cbx1^{-/-}$ mutation is fully penetrant with mice dying around birth possessing a variety of lesions including a severe genomic instability [47]. Disruption of the Cbx3 gene (encoding HP1γ) results in infertility and an increased postnatal mortality [50–52]. Consistent with the mutational analysis, unbiased exome data predict HP1β to have the highest probability of loss of function intolerance (pLI) out of the HP1 proteins[1] [53].

Mammalian HP1 proteins were some of the first proteins used in non-invasive fluorescence recovery after photobleaching (FRAP) studies to probe chromatin protein interactions in living cells [54]. Numerous FRAP studies, in conjunction with kinetic modelling, have shown that at steady-state equilibrium, the nuclear HP1 pool can be separated into three kinetic fractions: a highly mobile 'fast' fraction that freely diffuses through the nucleoplasm, a less mobile 'slow' fraction that binds to the HP1 ligand, H3K9me3 in heterochromatin, and a small immobile HP1 fraction whose ligand(s) is not known [55–57], although it has been suggested that this tightly bound fraction may involve the interaction of HP1 proteins with the histone H3 'core' [35,58]. These data have been interpreted as heterochromatin being a stable, membrane-less, nuclear compartment whose structural integrity is mediated by protein–protein and protein–RNA interactions, where the bulk of the constituents exchange freely with the surrounding nucleoplasm [59]. This view presages thermodynamic models of intracellular phase separation, which have led to physiochemical explanations for the biogenesis and maintenance of different nuclear compartments found in living cells (reviewed in [60–63]).

An exemplar of a nuclear compartment that is formed by liquid–liquid phase separation (LLPS) is the nucleolus, which is assembled at transcriptionally active ribosomal DNA loci [64,65]. Detailed examination of nucleoli has shown they are not composed of a single condensed phase surrounded by a dilute phase but consist of subcompartments where a secondary condensed phase is contained within the primary condensed phase; the subcompartments have distinct viscosities, surface tensions and protein compositions [66,67]. In vitro studies on mammalian HP1α have shown that HP1α undergoes LLPS and led to the suggestion that HP1α-dependent constitutive heterochromatin might also consist of phase-separated subcompartments; specifically a soluble phase, a liquid droplet phase and a gel-like phase [26,27]. The propensity to form liquid droplets in vitro is peculiar to HP1α because, under the same conditions, HP1β and γ do not form liquid droplets [68]; there seems not to be a relationship between the ability to form liquid droplets and pLI. On the face of it, the three phase-separated HP1α subcompartments appear to correspond with the three kinetic fractions observed in FRAP studies. However, a notable difference is that mammalian HP1α liquid droplets form in vitro independently of chromatin [25,27] while the 'slow' fraction is dependent upon the in vivo interaction of HP1 with its ligand, H3K9me3 [55,57]. Moreover, the CasDrop system has shown that mammalian HP1α is unlikely to form liquid droplets in vivo [69]. Instead, HP1α-rich foci co-localize with constitutive heterochromatin [29] rather than forming droplets that surround heterochromatin [69] indicating that if HP1α does drive phase separation, it does so in a manner different from multivalent intrinsically disordered proteins [62], which are known to form endogenous liquid–liquid phase-separated condensates in the nucleus [69]. These data indicate that phase separation of mammalian constitutive heterochromatin is unlikely to be mediated by HP1-driven LLPS, albeit work in Drosophila [70] and fission yeast [71] shows that HP1 proteins can drive LLPS. Rather, we suggest that the major mechanism by which mammalian HP1 proteins drive phase separation is polymer–polymer phase separation (PPPS; [72]) as opposed to LLPS.

Evidence that LLPS is unlikely to be the mechanism by which HP1 proteins form phase-separated compartments also comes from studies on their ability to act as bridging molecules between distant chromosomal loci. Expression of a lacI–HP1βCD fusion in a mouse cell line harbouring an approximately 10 kB lac operator (lacO; [73]) integrated into the telomeric end of chromosome 11 increased the frequency of contacts between the lacI–HP1βCD fusion bound to the lacO allele and H3K9me3-marked peri-centromeric heterochromatin [74]. The bridging effect requires the CD-H3K9me3 interaction because a point mutation (T51A in HP1βCD) that disrupts the interaction of the CD with H3K9me3 decreased the frequency of contacts to that observed for the wild-type (wt)

[1]pLI is a probability calculation of how well loss of function mutations can be tolerated, where 1.00 means the loss of function of the protein cannot be tolerated. HP1β holds a pLI of 0.95, while HP1α a lower pLI of 0.84, and HP1γ a pLI of 0.63.

allele. This ability of HP1 proteins to promote contact between distant chromosomal loci is conserved. Using the same *lacO* system, a transgenic fly was generated where *lacO* was integrated into the telomeric end of the X-chromosome [75]. Expression of a lacI–HP1a fusion led to a discrete lacI–HP1a signal at *lacO* on polytene chromosomes [75]. Assembly of lacI–HP1a at the *lacO* promoted chromosome folding and association of the lacI–HP1a–*lacO* complex with loci at distant chromosomal sites. Bridging between two loci was dependent upon HP1a CSD dimerization. The bridging effect in transgenic flies was all the more impressive given it was observed in polytene chromosomes, which do not fold easily because of their stiffness that is a consequence of the approximately 1000 sister chromonemata that align in complete register, resulting in the giant cross-banded chromosomes [76]; reduced flexibility (increased stiffness) is the reason given for the absence of compartmental domains in Hi-C maps generated from polytene nuclei [77]. It is unlikely that HP1 liquid droplets could mediate this long-range bridging effect. For one, a liquid droplet would surround chromatin, whereas the lacI–HP1a forms a tightly localized domain at *lacO* [75]. Further, the bridging effects observed *in vitro* and *in vivo* are mediated by direct stereospecific interactions requiring the modular domains of HP1 [74,75,78], whereas liquid droplets pull genomic regions together by proxy *via* surface tension driven coalescence [69].

The 11 nm nucleosome fibre is a polymer [79,80]. The physics of polymers is well understood and predicts that very small interactions between monomers can strongly influence the whole structure because many small interactions can add up to stabilize different structures [81,82]. For example, when a homo-polymer is placed in a solvent, the interaction of monomers with themselves and the solvent can lead to an incompatibility that results in phase separation of the polymer (i.e. the polymer de-mixes because the energetic cost of mixing the polymer in the solvent is prohibitive), whereupon the polymer collapses into structures such as ordered globules surrounded by a solvent-rich phase [81–85]. Notably, the formation of ordered globules is likely to be one of the consequences of folding of the nucleosome fibre 'polymer' in the nucleus. This was concluded from the first Hi-C study, which showed that the average contact probability $P_c(s)$ between two loci at a distance $s$ is a power law, $P_c(s) \sim 1/s^\alpha$, having a specific exponent $\alpha = 1.08$ for genomic distances up to the size of several megabasepairs [86]. This observation led to the notion that the nucleosome fibre adopts a specific state found in ideal polymer chains models, namely the ordered (fractal) globule, which has an exponent $\alpha = 1$ [85]. Scaling of $s^{-1}$ is best understood in terms of the nucleosome fibre folding into small 'globular' regions which condense into larger globules that then form even larger globules [85] (figure 6). The resulting structure is self-similar (i.e. fractal) over two orders of magnitude from around the level of whole chromosomes down to a scale of a few hundred kilobasepairs [86]. Subsequent Hi-C studies have shown that the exponent can vary depending on chromosome and cell type, indicating that the nucleus contains a complex mixture of differently folded regions, including ordered (fractal) globules, controlled by basic mechanisms of polymer physics that are strongly influenced by chromatin binding proteins and epigenetic modifications [87].

When homo-polymers in solution are cross-linked, they can form condensed structures that are incompatible with the surrounding solvent, resulting in de-mixing and phase separation [88,89]. Highly cross-linked chromatin, such as that found in mitotic chromosomes, forms phase-separated polymer gels [90]. Similarly, phase separation could arise when H3K9me2/3-marked nucleosome 'polymers' are 'cross-linked' by HP1-mediated bridging within and between the 'polymer' fibres. Bridging by HP1 proteins has been definitively demonstrated through elucidation of the three-dimensional structure of H3K9me3-containing di-nucleosomes complexed with human HP1α, β or γ [78] (figure 2). Two H3K9me3 nucleosomes are bridged by a symmetric HP1 dimer *via* the H3K9me3-CD interaction; the linker DNA between the nucleosomes does not directly interact with HP1 [78]. *In vivo* FRAP studies indicate that the binding of HP1 to H3K9me3-marked nucleosomes is dynamic and transient, but the rapid and constant exchange with unbound HP1 in the nucleoplasm [55–57] would maintain the bridging interactions. Given the rapid turnover of H3K9me3-bound HP1, HP1-mediated bridging may stabilize or promote pre-existing (condensed) chromatin compaction states rather than inducing these states de novo. Notably, the interaction of HP1α and β with H3K9me3-marked nucleosomal fibres revealed that HP1 dimers can bridge different segments of the same fibre [31,75]. HP1 dimers can also bridge between H3K9me3-marked nucleosomal fibres, i.e. drive inter-fibre interactions [31,75]. Given the flexible HR, HP1-bridging could promote or stabilize the conformation of H3K9me3-marked nucleosome fibres, within and between larger 'clutches' of nucleosomes that are known to be enriched in heterochromatin [5,6] (figure 3). Together with osmotic depletion attraction [7], the HP1-mediated bridging between 'clutches' would promote their coalescence with the potential to form/stabilize (fractal) globules (figure 3) that could contribute to PPPS of constitutive heterochromatin.

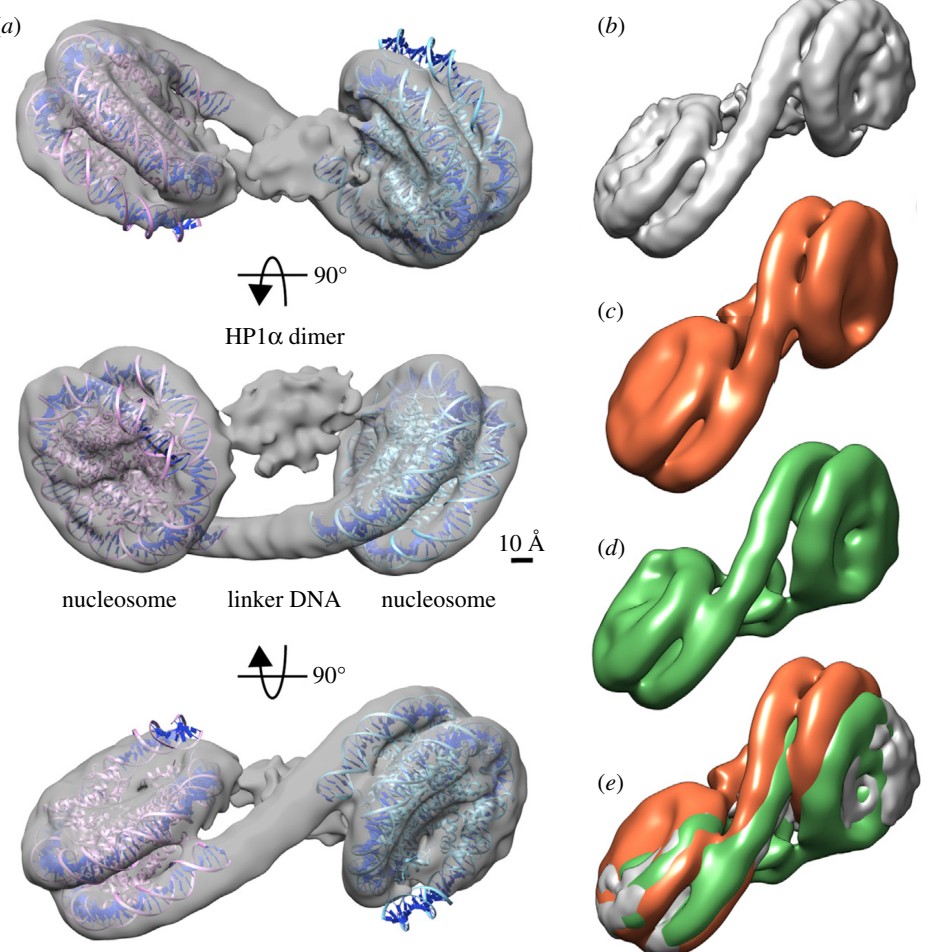

**Figure 2.** Bridging of two nucleosomes by HP1 as seen in the structure of the HP1α/β/γ-dinucleosome complex. In (*a*), three orthogonal views of the reconstructed three-dimensional structure of HP1α–dinucleosome complex. A model of the nucleosome core particle (PDB code 3LZ0) is docked into each of the two nucleosome densities. The linker DNA and the bridging HP1α dimer are clearly distinguishable and the linker does not interact with the HP1α dimer. Scale bar, 10 A°. In (*b*)–(*d*), the HP1α–dinucleosome complex (*b*), the HP1β–dinucleosome complex (*c*) and the HP1γ-di-nucleosome complex (*d*) are shown. In (*e*), all three HP1–dinucleosome complexes are superimposed, where the HP1α–dinucleosome complex is in grey, the HP1β–dinucleosome complex is in orange and the HP1γ–dinucleosome complex is in green. The orientations of the left nucleosomes are fixed. Taken from [78].

# 3. Constitutive heterochromatin, heterochromatin-*like* domains and complexes

Constitutive heterochromatin is found at distinct chromosomal territories—around the centromeres (peri-centromeric), at the telomeres and flanking the peri-nucleolar regions (for reviews, see [91–94]). They are huge, ranging up to 20 Mb in size in mouse and man (table 1). Their size makes them cytologically visible, which enabled Heitz [1] to demonstrate graphically compartmentalization of the eukaryotic genome. In mammals, the bulk surrounds the centromeres (peri-centromeric) and, surprisingly, there is no conserved sequence one can point to that is known to cause nucleation at this site. Instead, it is thought the generally repetitious nature of sequences promotes nucleation of peri-centromeric constitutive heterochromatin [91]. In addition to repetitious DNA, there are proteins, RNAs and epigenetic modifications enriched within constitutive heterochromatin [163,164] (table 1) that are thought to be involved in its nucleation, assembly and propagation [91]. Several of these constituents may contribute to PPPS of constitutive heterochromatin the most likely, but not exclusively, being affinity of homotypic DNA repetitive elements for each other [165], mutual affinity of nucleosomes that share the same modified histones and proteins that bridge between DNA and nucleosome fibres. Along with HP1 proteins, bridging is a property shared by many other proteins

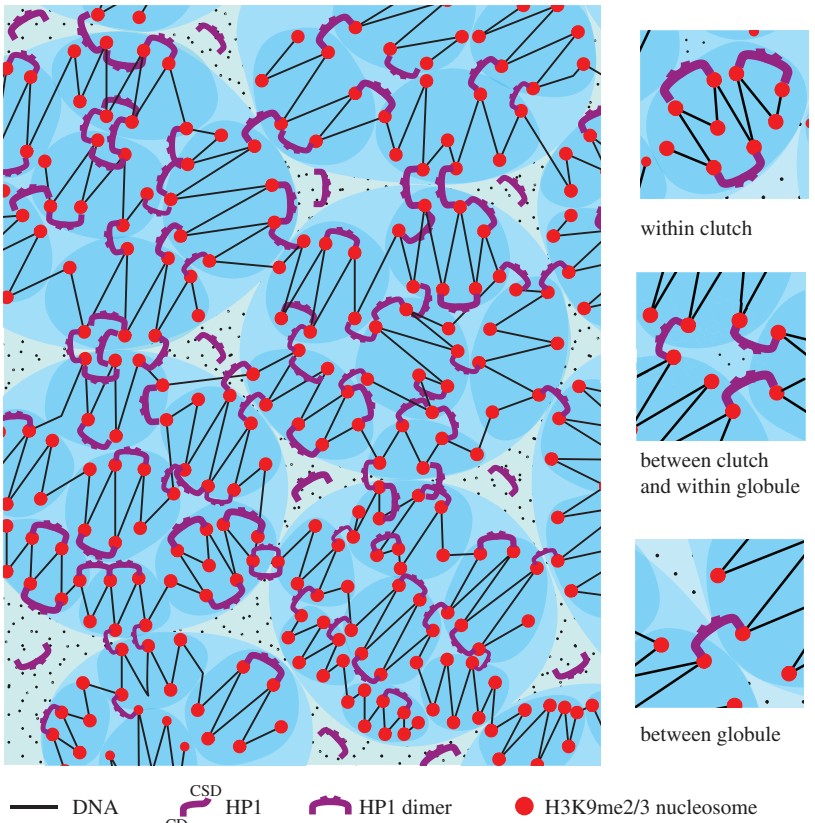

within clutch

between clutch
and within globule

between globule

DNA CSD CD HP1 HP1 dimer ● H3K9me2/3 nucleosome

**Figure 3.** A schematic model for how HP1 bridging of H3K9me2/3-marked nucleosomes could contribute to PPPS of constitutive heterochromatin. 'Clutches' (smaller, darker, blue ovals) of H3K9me2/3-marked nucleosomes (red spheres) organized in an irregular zig-zag structure where linker length is variable. Given the zig-zag organization, HP1 proteins preferentially 'bridge' nucleosomes that are second-nearest neighbours thereby stabilizing the zig-zag geometry (top inset on right). There is also bridging between 'clutches' (middle inset on right) that could, in addition to osmotic depletion attraction, result in merging of 'clutches' into larger globules (larger, lighter, blue ovals). Further bridging between globules (bottom inset on right) could lead to the formation of self-similar larger globules. These may then form even larger globules ('globules of globules'; not shown) [86]. The dark 'dots' represent molecular species that contribute to the entropic molecular crowding effect that promotes merging.

that possess two (or more) chromatin/DNA-binding motifs [166,167], any of which could contribute to PPPS of constitutive heterochromatin. For example, methyl binding proteins might contribute by bridging methylated nucleosomal DNA fibres [3]. Putting it short, there are several factors that are likely to contribute to PPPS of cytologically visible constitutive heterochromatin, with the hallmarks HP1 and H3K9me2/3 being the most conserved.

The demonstration that genes encoding HP1 proteins are highly conserved was accompanied by the prediction that HP1-containing heterochromatin-*like domains* and *complexes* would exist *outside* canonical constitutively heterochromatic territories and regulate the cell-to-cell (epigenetic) inheritance of chromatin states [168] (for reviews, see [28,35,95,169]). Many such domains and complexes that share structural components (e.g. HP1) and epigenetic modifications (e.g. H3K9me3) have now been identified (table 1). An attempt at a rough classification as a domain or complex has been made on the basis of size [95], with *domains* being in the region of 0.1–5Mb in size (table 1). Included in the *domains* are the odorant receptors [133,134,170], the Krüppel-associated box (KRAB) domain zinc finger (KRAB-ZNF) gene clusters [138] contained within the B4 subcompartment [140]; the protocadherin topologically associated domain (TAD; [142]), somatic cell nuclear transfer (SCNT) reprogramming resistant regions (RRRs; [141]) and the Zscan4 gene cluster [171]. There is evidence that these domains assemble regions of chromosomal DNA involved in regulating cell fate [143,144]. Moreover, perturbation in their assembly and propagation is likely to affect cellular identity [172,173] and SCNT reprogramming efficiency [141]. Heterochromatin-*like complexes* are small, less than 100 kb and usually only a few kb in size (table 1). They include the 3' end of the KRAB-ZNF genes [138,139], imprinted

**Table 1.** Major structural and enzymatic constituents, along with histone/DNA modifications, associated with mammalian constitutive heterochromatin, which are shared with heterochromatin-like domains/complexes. (✓, present; NK, not known. References are in square brackets and are to be found in the reference list. Taken and modified from [95].)

| | size | H3K9MTases | H3K9me3 | H4K20MTase | H4K20me3 | HP1 | DNMTases | SmC | Np95 | ATRX/DAXX | H3.3 | KAP1 | compartment/TAD |
|---|---|---|---|---|---|---|---|---|---|---|---|---|---|
| **constitutive heterochromatins** | | | | | | | | | | | | | |
| peri-centric | human ~0.2–20 Mb [96]; mouse ~6 Mb [97] | ✓ SUV39H1/2 [98,99] G9a/ GLP [100] SETDB1 [101] | ✓ [102] | ✓ SuvH4201/2 [103] | ✓ [29,103] | ✓ HP1α [29,104] HP1β [29,105] HP1γ [29,106] | ✓ DNMT1 [107] DNMT3A [108] DNMT3B [109] | ✓ [110] | ✓ [111] | ✓ [112] | ✓ [113–115] | ✓ [101] | PAD identified by 4C not Hi-C [74] |
| telomeric plus sub-telomeric | human ~10–300 kb [96]; mouse ~5 Mb [116] | ✓ SUV39H1/2 [117] SETDB1 [118] | ✓ [117,119] | ✓ SuvH4201/2 [119,120] | ✓ [119–121] | ✓ HP1α HP1β [119–121] | ✓ DNMT1 DNMT3A DNMT3B [119–121] | ✓ [119–121] | NK | ✓ [113–115] | ✓ [113–115] | NK | TPE-OLD [122] |
| NOR plus peri-nucleolar | human ~0.25–6.5 Mb [92,123]; mouse NK | ✓ SUV39H1 [124] | ✓ [125,126] | ✓ SuvH420/2 [127] | ✓ [126] | ✓ HP1α HP1β HP1γ [128] | ✓ DNMT1 DNMT3A [129,130] | ✓ [129,130] | NK | ✓ [131] | ✓ [131] | NK | NAD [132] |
| **heterochromatin-like domains** | | | | | | | | | | | | | |
| odorant receptors | human 0.1–1 Mb [133] mouse 1–5 Mb [134] | ✓ G9a/GLP [135] | ✓ [135,136] | ✓ SuvH4201/2 [135] | ✓ [135,136] | ✓ HP1β [137] | NK | NK | NK | NK | NK | NK | predominantly B2 [this study] |
| KRAB-ZNF gene clusters | human and mouse up to 4 Mb [138] | ✓ SUV39H1 [138] | ✓ [138,139] | NK | NK | ✓ HP1β [138,139] | NK | NK | NK | NK | NK | ✓ [139] | B4 sub-compartment [140] |
| SCNT reprogramming-resistant regions (RRRs) | human NK; mouse up to 2 Mb [141] | ✓ SUV39H1 [141] | ✓ [141] | NK | NK | NK | NK | NK | NK | NK | NK | NK | mixed A and B [this study] |
| protocadherin cluster superTAD | human and mouse 1.2 Mb [142] | ✓ SETDB1 [142] | ✓ [142] | NK | NK | NK | NK | NK | NK | NK | NK | NK | superTAD [142] B3 sub-compartment [this study] |
| sonication-resistant heterochromatin (srHC) | human average size 0.135 Mb [143] | NK | ✓ [143,144] | | NK | NK | NK | NK | NK | NK | NK | NK | predominantly B2 [this study] |
| **heterochromatin-like complexes** | | | | | | | | | | | | | |
| 3' end of KRAB-ZNF genes | human and mouse ~6 kb [138] | ✓ SETDB1 [145,146] | ✓ [145,147] | NK | NK | ✓ HP1β [138,139] | NK | NK | NK | ✓ [147] | ✓ [147] | ✓ [145,148,149] | B4 sub-compartment [140] |
| IPS reprogramming-resistant regions | human NK; mouse NK [150,151] | ✓ SETDB1 [150,151] | ✓ [150–152] | NK | NK | ✓ HP1γ [152] | NK | NK | NK | NK | NK | NK | NK |
| imprinted gDMRs | human NK; mouse ~6 kb [153] | ✓ SETDB1 [154] | ✓ [153–155] | ✓ SuvH4201/2 [156] | ✓ [153,156,158] | ✓ HP1α HP1β [153] HP1γ [153,155,157] | ✓ DNMT1 DNMT3A DNMT3B [157,158] | ✓ [159,160] | ✓ [158] | ✓ [161] | ✓ [161] | ✓ [157,162] | mixed: A and B [this study] |

gDMRs [153] and the SETDB1-regulated iPS reprogramming resistant regions [150]. The targeted assembly of these domains/complexes contributes to the coarse-grained chromatin-state pattern that characterizes mammalian genomes [174,175].

There are four different but related questions that need to be addressed in order to understand how heterochromatin-*like* domains/complexes regulate chromatin-templated processes and genome organization. First, how is the assembly of a heterochromatin-like domain/complex nucleated at a particular site in the genome? Second, how does a larger domain form by spreading along the nucleosome fibre from that site? Third, how is the domain/complex epigenetically inherited from one cellular generation to the next? Finally, how do such domains/complexes contribute to compartmentalization of the genome in terms of heterochromatin and euchromatin? Clues to what the answers might look like have come from studies on the heterochromatin-*like* domains that encompass the KRAB-ZNF gene clusters (table 1) [138]. In humans, the majority of clusters reside on chromosome 19. The KRAB-ZNF genes encode the largest family of transcriptional regulators in higher vertebrates [176] and the general features of the heterochromatin-*like* domains that encompass the KRAB-ZNF genes clusters are well described. The mammalian HP1 protein, HP1β, and the K9 HMTase SUV39H1 are enriched at the KRAB-ZNF gene clusters and the 20 domains on human chromosome 19 range from 0.1 to 4 Mb in size [138]. HP1β binding is elevated throughout the clusters compared to regions outside the clusters and high-resolution analysis of a specific cluster on chromosome 19, encompassing the ZNF77 and ZNF57 genes, has shown that HP1β binding is co-extensive with H3K9me3 [139]. The formation of large heterochromatin-*like* domains that encompass the clusters is thought to 'protect' the KRAB-ZNF gene repeats as they have expanded during evolution by preventing illegitimate recombination [138], rather than to repress and silence the KRAB-ZNF genes [139,145]. Notably, there are significant variations along a cluster, with enrichment of HP1β at the 3′ end of KRAB-ZNF genes and depletion in the 5′ promoter regions [138,145]. The enrichment observed at the 3′ end has focused attention on the molecular mechanism by which a heterochromatin-*like* domain can be nucleated, which brings us to the first of four questions that need to be addressed.

## 3.1. A localized heterochromatin-like complex nucleates formation of the larger domain

Nucleation of the domain probably involves the assembly of a specific, localized, heterochromatin-*like* complex at the 3′ end of the KRAB-ZNF genes (table 1 and figure 4*a*). Nucleation is necessarily sequence-specific and intriguingly enough requires binding of a sequence-specific KRAB domain-zinc finger protein (KRAB-ZFP) to the 3′ end of the KRAB-ZNF genes [146,148]. Once bound to its cognate recognition sequence, the KRAB-ZNP recruits the universal co-repressor of KRAB-ZFPs, KRAB-associated protein 1 (KAP1; also known as Tif1β, TRIM28 or KRIP1) [181,184,185]; the KRAB motif of the DNA-bound KRAB-ZFP binds to the RBCC domain of KAP1 [148]. KAP1 acts as a 'scaffold' for different enzymatic and structural components that are essential for the nucleation process (figure 4*a*). A key interaction is that of the SETDB1 HMTase with KAP1 that leads to the generation of the H3K9me3 modification [145,146]. SETDB1 binds the sumoylated form of the KAP1 bromodomain; the sumoylated version is the active, most repressive, form of KAP1 [178]. Sumoylation is mediated intra-molecularly— the KAP1 PHD domain is an E3 ligase that cooperates with UBE2i (also known as UBC9) to transfer SUMO2 [179] to the KAP1 bromodomain. KAP1 also recruits a dimer of HP1 molecules through the PxVxL motif in KAP-1 called the HP1-box [186,187]; KAP1 binds equally well to all three HP1α/β/γ isotypes in biochemical assays [186,187]. Nucleation is reinforced by a specific mechanism that continually replenishes the repressive H3K9me3 modification. Specifically, KAP1 binds to DAXX [188], which is an H3.3-specific chaperone [113–115] that incorporates H3.3 at the 3′ end of the KRAB-ZNF genes [147], whereupon H3.3 is tri-methylated at K9 by SETDB1 [146]. The binding of the ATRX–DAAX complex is enhanced by the known interaction of ATRX with both H3K9me3 and HP1, the former through the ADD domain and the latter through an LxVxL motif, and both interfaces combine to localize ATRX to heterochromatin [180]. The SETDB1–HP1–ATRX complex is stable *in vivo*: when HP1 is artificially repositioned within the nucleus, both SETDB1 and ATRX are relocated along with it [189].

Several additional enzymatic activities likely to be part of the nucleation process have been revealed using artificially reconstituted systems where a regulatable KRAB domain is targeted to a synthetic sequence that drives a reporter gene [149,190–192]. The small (approx. 1.5 kB) heterochromatin-*like* complexes generated by the targeted KRAB domains possess elevated levels of both H4K20me3 and H3K9me3 [191,192], indicating the recruitment of a H4K20me3 HMTase and operation of the H3K9me3:HP1:H4K20me3 pathway [182]. The KAP1 recruited by the KRAB domain also binds the NuRD complex that deacetylates histones [190]. There are also increased levels of DNA methylation

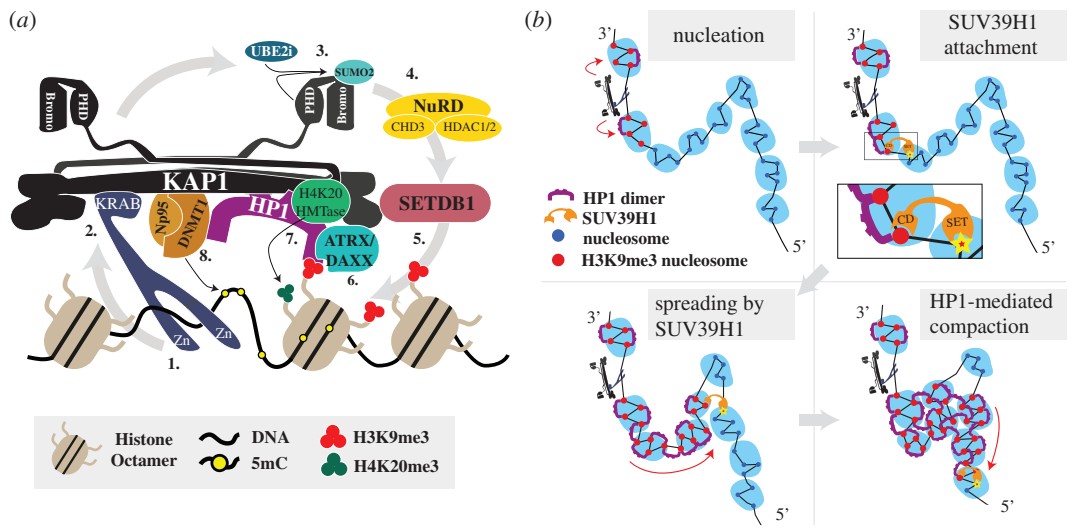

**Figure 4.** Nucleation and spreading of a KRAB-ZNF heterochromatin-*like* domain. In (*a*) is the heterochromatin-*like* complex that is assembled at the 3′ end of KRAB-ZNF genes. These complexes nucleate the KRAB-ZNF heterochromatin-*like* domains that encompass the KRAB-ZNF gene clusters. The diagram is based on the KAP1 and HP1 interactomes. (**1**) The KRAB-ZFP binds to its DNA-binding site through its zinc-fingers (Zn). (**2**) The KRAB domain of the KRAB-ZFP interacts with the RBCC domain of KAP1 [146]. The structure of KAP1 is taken from [177]. An HP1 CSD dimer binds to one molecule of KAP1 through the PxVxL motif (the HP1-box). The HP1 CD binds to H3K9me3. (**3**) The PHD domain of KAP1 is an E3 ligase that cooperates with UBE2i to sumoylate the KAP1 bromodomain [178,179]. (**4**) The sumoylated bromodomain is bound by the NuRD complex that deacetylates acetylated histones (green circle) in preparation for histone methylation [180]. (**5**) SETDB1 H3K9 HMTase interacts with the sumoylated bromodomain [181] and generates H3K9me3 (orange circles). (**6**) The ATRX/DAXX complex is bound to KAP1, HP1 and H3K9me3. ATRX/DAXX incorporates replacement histone H3.3 into chromatin thereby reinforcing nucleation [147]. (**7**) HP1 recruits a H4K20 HMTase that generates H4K20me3 (orange circles). This is the H3K9me3:HP1:H4K20me3pathway. (**8**) The maintenance DNA methylase DNMT1 binds to KAP1 [157,158]. Np95 is the cofactor of DNMT1 and is also recruited by KAP1 [182]. DNMT1 maintains cytosine methylation at the site of assembly [157,158]. Not shown are the the de novo DNMTases, DNMT3A and DNMT3B, which can interact with KAP1 [157,158]. It is known that the H3K9me3 and HP1 enrichment at the 3′ end of the KRAB-ZNF genes extend approximately 6 kb at the site of assembly of the nucleation complex [138,139] (table 1). Taken and modified from [95]. In (*b*) is a coarse-grained model depicting SUV39H1-mediated spreading of H3K9me3 and HP1 proteins that form the larger KRAB-ZNF domain. The top left-hand panel depicts the nucleation complex (shown in *a*) generating H3K9me3-marked nucleosomes (red filled circles) in 'clutches' on either side of the complex. The top right-hand panel shows the CD of SUV39H1 attaching to H3K9me3-marked nucleosome within a clutch whereupon the SUV39H1 SET domain catalyses the SAM-dependent methylation of H3K9, which, therefore, provides a positive feedback loop [183] that enables spreading of the domain in the 5′ direction away from the nucleation site [139]. In the bottom left-hand panel, SUV39H1 is depicted as mediating 'looping-driven' spreading of H3K9me3 (red arrow), where spreading is not restricted to next-neighbour interactions but can skip nucleosomes, as it does here where one 'clutch' is 'looped-out' and whose constituent nucleosomes are not methylated at H3K9. This is likely to be relevant to the KRAB-ZNF genes which show a depletion of H3K9me3 and HP1β at their 5′ ends [138,139]. In the wake of the newly deposited H3K9me3 HP1 dimers bind H3K9me3 through their CDs. The bottom right-hand panel depicts continued spreading of H3K9me3 by SUV39H1 activity (red arrow) and HP1-mediated bridging of H3K9me3-marked nucleosomes within and between 'clutches'. Bridging results in chromatin compaction.

[191,193,194], which is consistent with biochemical assays showing that KAP1 binds to all three DNA methyltransferases and the DNMT1 cofactor Np95 [157,158]; HP1 also interacts with all three DNA methyltransferases [195,196].

## 3.2. Spreading to form a heterochromatin-like domain

Once a heterochromatin-*like* complex is nucleated at the 3′ end of the KRAB-ZNF genes, a larger domain is generated by 'spreading' from that site through the activity of the SUV39H1 HMTase [138]. Spreading moves away from the nucleation site towards the 5′ end of the genes [139]. SUV39H1 is the archetypal H3K9HMTase that generates the H3K9me3 to which HP1 binds [24]. The most recent model posited for SUV39H1-mediated spreading involves a 'two-step' activation of SUV39H1 [183]. First, a highly mobile SUV39H1 with low HMTase activity attaches via its CD to H3K9me2/3 (figure 4*b*, second

panel) that would be generated by the nucleation complex (figure 4b, first panel). The second step involves H3K9 methylation of adjacent nucleosomes owing to enhanced HMTase activity of the 'anchored' SUV39H1 (figure 4b, second panel). This mechanism is self-reinforcing because the SUV39H1CD binds newly methylated H3K9 and in this way re-iterates along the nucleosome fibre, whereupon HP1 binds in its wake to the H3K9me3-marked nucleosomes (figure 4b, third panel). The model is a refinement of earlier models where spreading involves a known interaction of SUV39H1 with HP1 [197,198], where it is the CD of HP1 that binds newly methylated H3K9-nucleosomes and recruits SUV39H1 to reinforce and continue spreading [37]. HMTase-generated H3K9me3 'spreads' at the rate of approximately 0.18 nucleosomes hr$^{-1}$ [199]. The precise character of SUV39H1-mediated 'spreading' has not been determined, but it may be linear or involve a looping-driven propagation [183]. We have depicted 'spreading' as a looping-driven propagation (figure 4b, third panel) in order to accommodate the 'skipping' of the 5′ end of the KRAB-ZNF genes where HP1β is depleted [138]. Neither is it known how the spreading and thus domain size of the KRAB-ZNF clusters is limited, although several (boundary) sequence elements and associated proteins have been documented that can modulate the size and shape of heterochromatin domains [200,201].

## 3.3. Genomic bookmarking and epigenetic inheritance

During mitosis, HP1 proteins are removed from chromatin, only to re-associate in the following interphase [202]. Consequently, in order for a heterochromatin-*like* domain to be epigenetically inherited from one cell generation to the next, the site at which it is nucleated in the genome must be 'bookmarked' so that it is re-nucleated at that specific site after mitosis. Nucleation complexes (figure 4a) are excellent candidates for genomic 'bookmarks'. They are assembled at specific sites through the binding of KRAB-ZFPs to their cognate recognition sequences (figure 4a). The heterochromatin-like complex so targeted nucleates the subsequent 'spreading' and formation of a larger domain (figure 4b). A coarse-grained polymer model of genomic bookmarking predicts, as one of three parameters required for robust epigenetic inheritance of chromatin states, a critical density of bookmarks along a chromatin fibre to be 1 or 10 nucleosomes per 400 nucleosomes ($\phi_c \sim 0.04$; [203]). We have made a rough estimate of the density of nucleation sites within the B4 subcompartment using the distribution of KAP1 peaks (figure 5b–d). In the B4 sub-compartment (14 642 kb), there are conservatively 353 KAP1 peaks, which corresponds to one nucleation site per 40 kb, i.e. 200 nucleosomes, where 1 nucleosome is 200 bp. The base of the peak of enrichment for H3K9me3 and HP1β around the KAP1sites gives the size of the nucleation site at approximately 6 kb (figure 5d), which is in agreement with previous studies [138,153]. This indicates a density of bookmarks of approximately 30 nucleosomes per 200 nucleosomes in the B4 subcompartment, well within the critical density defined by the coarse-grained polymer model. The second parameter is that bookmarking involves the sequence-specific recruitment of the machinery that epigenetically modifies chromatin. The KRAB-ZFP that targets the nucleation complex (figure 4a) satisfies that requirement. The third parameter is the operation of a positive feedback mechanism that can spread and establish the domain. This too is met because of the self-reinforcing SUV39H1-mediated spreading (figure 4b) that generates the domain from the nucleation site (figure 4a). The organization of the KRAB-ZNF heterochromatin-*like* domains therefore satisfies the theoretical requirements for 'epigenetic domains' [203].

Replication of the heterochromatin-*like* domains/complexes during S-phase has been treated in detail elsewhere [95]. Briefly, it involves two complexes called the CAF-1 and SMARCAD1 complexes; both complexes contain KAP1 and HP1 and K9 HTMases that are key components involved in the replication of heterochromatin [100,101,204,205].

## 3.4. Heterochromatin-*like* domains/complexes and compartmentalization

Contacts of heterochromatin-*like* domains/complexes with constitutive heterochromatin are likely to lead to coalescence and promote *macroscopic* PPPS. When heterochromatin-*like* domains/complexes are brought into contact with constitutive heterochromatin, where the latter is enriched in high concentrations in H3K9me3, many HP1–H3K9me3 inter-fibre contacts will be formed [31] (figure 6a). The domains/complexes will merge with the large blocks of constitutive heterochromatin because their constituents are, by definition, essentially the same (table 1) and bridging molecules, such as HP1, are unable to distinguish between the translocated domain/complex and the large block of heterochromatin. Bridging and merging of the domain/complex with constitutive heterochromatin would contribute to de-mixing and macroscopic phase separation [88,89] (figures 5a and 6a), with domains/complexes becoming seamlessly part of micrometre-sized cytologically visible constitutive

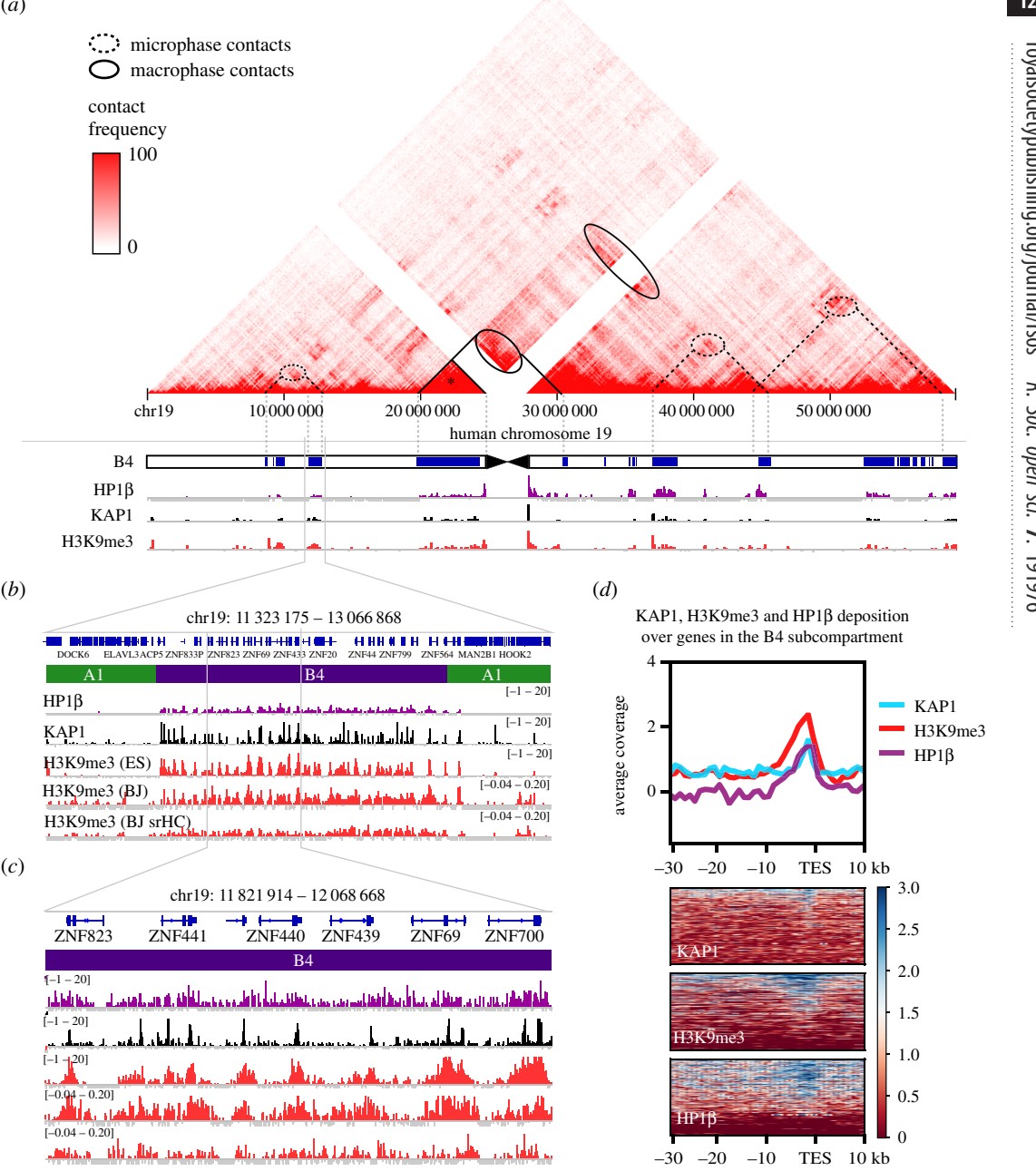

**Figure 5.** The B4 subcompartment and the density of nucleation sites within the subcompartment. (*a*) The contact frequency map in H1-ESCs from which the B4 subcompartment emerges. The B4 subcompartment (annotation combined from [138] and [140]) overlaps exactly with regions enriched for HP1β, KAP1 and H3K9me3, which are given below the chromosomal map. Increased contact frequency can be observed within peri-centric regions, such as that marked with the asterisk (*), which denotes a large peri-centric KRAB-ZNF cluster that probably undergoes macro-phase separation (see figure 6 and text for details). Contacts between the peri-centric KRAB-ZNF cluster and more distal sites (ovals demarked by solid lines) are also likely to undergo marco-phase separation. Also shown are contact enrichments between non-peri-centric B4 subcompartments (ovals with dotted lines) that represent contacts between micro-phase-separated HP1-containing block co-polymers (see figure 6 and text for details). The magnifications in (*b*) and (*c*) illustrate the overlap of the B4 compartment with KAP1/H3K9me3/HP1β in more detail. Nucleation sites, as defined by KAP1 peaks, occur at a frequency of 1 every approximately 40 kb over the B4 subcompartment. (*b*,*c*) The distribution of H3K9me3 and sonication-resistant heterochromatin (srHC) within the B4 subcompartment, which overlap with the other peaks. Tracks in (*b*) and (*c*) are input-subtracted ChIP of HP1β in HEK293, KAP1 in H1 ES cells, H3K9me3 in H1 ES cells, H3K9me3 from BJ cells, and H3K9me3 from BJ srHC, respectively. (*d*) The profile of KAP1, H3K9me3 and HP1β binding at the 3′ end of KRAB-ZNF genes, anchored by the transcription end site (TES). The KAP1 peaks are on average 1 kb in width and surrounded by HP1 and H3K9me3 enrichment that extend approximately 6 kb. Average coverage in (*d*) is input-subtracted ChIP reads per genomic context (RPGC) per 1000 bp bin.

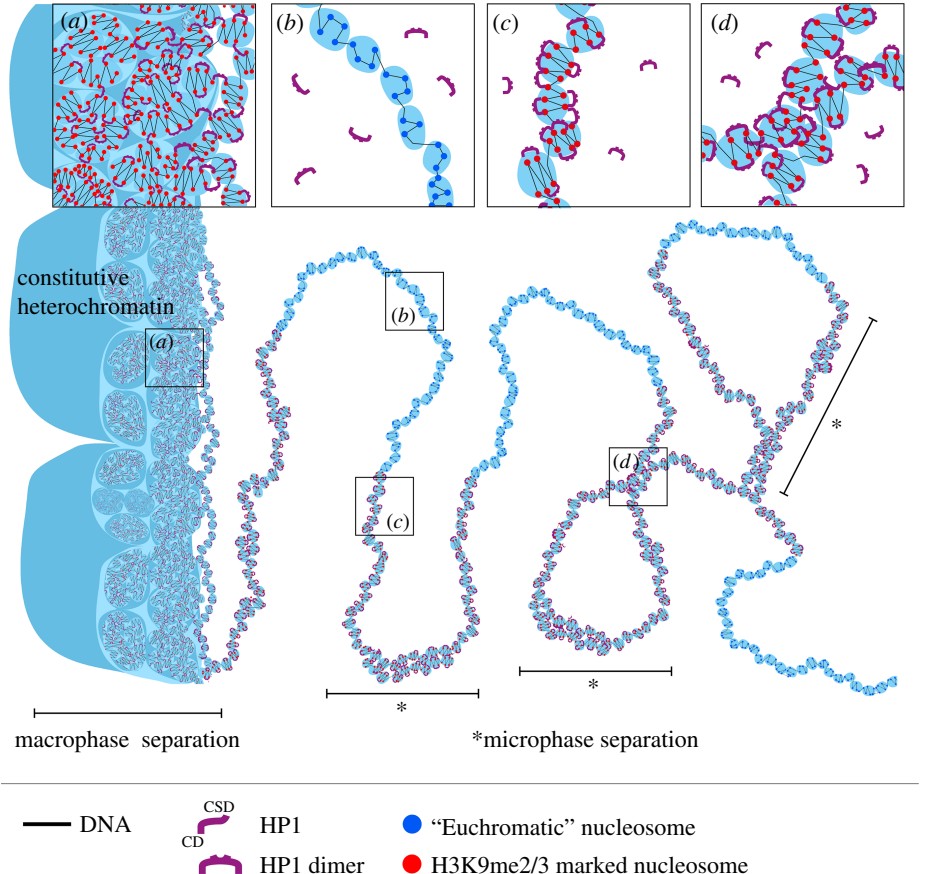

**Figure 6.** *Macro-* and *micro-*phase separation of heterochromatin-*like* domains/complexes. To the left of the figure, constitutive heterochromatin is depicted as ordered globules, ranging from smaller to larger. (*a*) 'Clutches' that contain H3K9me2/3-marked nucleosomes (red spheres) organized in an irregular two-start zig-zag organization. Within the 'clutches', HP1 proteins preferentially 'bridge' nucleosomes that are second-nearest neighbours in the zig-zag rather than closest neighbours, thereby stabilizing the zig-zag geometry. HP1 proteins also act as bridges within and between 'clutches' and 'globules'. Also shown in (*a*) is a heterochromatin-*like* domain consisting of 'clutches' of HP1-bridged H3K9me2/3-marked nucleosomes (red spheres) that runs alongside constitutive heterochromatin. When brought into close apposition to constitutive heterochromatin, an environment rich in HP1 and H3K9me3, extensive 'bridging' of the heterochromatin-*like* domain to constitutive heterochromatin takes place mediated by HP1 proteins. This contributes to *macro-*phase separation, where the heterochromatin-*like* domain merges with cytological-visible constitutive heterochromatin. The same heterochromatic fibre extends to the right into the nucleoplasm away from constitutive heterochromatin and joins a euchromatic segment consisting of 'clutches' containing 'euchromatic' nucleosomes (blue spheres in (*b*)) that exhibit less of the two-start contact geometry of H3K9me2/3-marked nucleosomes but rather possess a more disordered or heterogeneous organization. The 'euchromatic' fibre in turn extends into a heterochromatin-like domain/complex consisting of 'clutches' containing HP1-bridged H3K9me2/3-marked nucleosomes, as shown in (*c*). HP1-mediated bridging of H3K9me3-marked nucleosomes leads to enrichment of both and promotes *micro-*phase separation that is characteristic of BCPs (see text for details). 'Blocks' of heterochromatin-*like* domains/complexes can engage in *cis-* (shown in (*d*)) and *trans-*interactions (not shown). *Cis-*interactions, as shown in (*d*), could explain the emergence of the B4 subcompartment from contacts between the KRAB-ZNF heterochromatin-*like* domains on human chromosome 19.

heterochromatin. The ZNF91 KRAB-ZNF cluster that lies within peri-centric heterochromatin [206] most probably undergoes macroscopic phase separation (asterisk in figure 5*a*) and its interaction with more distal sites will also lead to macro-phase separation (ovals with solid lines in figure 5*a*). Macro-phase separation may also take place with odorant receptor genes that form very large heterochromatin-*like* domains up to 5 Mb in size (table 1), 45–50% of which exhibit a preferential localization to constitutive heterochromatin in post-mitotic olfactory sensory neurons [207]. Other KRAB-ZNF clusters spread out along the arm of chromosome 19 [138] are assembled into heterochromatin-*like* domains where *cis-* contacts between them (ovals with dotted lines in figure 5*a*) contribute to the B4 heterochromatic sub-compartment in Hi-C experiments [140].

Heterochromatin-*like* domains along the arms of chromosome 19 that are flanked by stretches of 'euchromatic' nucleosomes (figure 5*a*) will, we suggest, behave like blocks in a block copolymer (BCP). Polymers that contain blocks of at least two (or more) different types of monomer are called BCPs [208,209], where a block is made up of identical monomers. BCPs share many of the properties of homo-polymers excepting that the covalent bond connecting the different types of block prevents *macroscopic* phase separation. Instead, BCPs undergo *microscopic* phase separation when one block becomes highly enriched and incompatible resulting in phase separation to form nanostructures in the range of 0.1–100 nm [209]. For a heterochromatin-*like* 'block', enrichment (leading to incompatibility) would result from HP1-mediated bridging of H3K9me2/3-marked nucleosomes *within* the 'block' (figure 6*c*) and this could drive micro-phase separation (asterisks in figure 6). *cis*- contacts mediated by HP1 'bridging' *between* micro-phase-separated heterochromatin-*like* 'blocks' (figure 6*d*; ovals with dotted lines in figure 5*a*) could explain the emergence of the B4 compartmental domain identified in Hi-C maps [140] (figure 5*a*). Specifically, given that: (i) KRAB–ZNF clusters are assembled into heterochromatin-*like* domain (blocks) enriched in H3K9me3 and HP1 [138,139], (ii) HP1 can act as a 'bridge' between H3K9me3-marked nucleosomes [31,75,78], and (iii) HP1 can also act as a bridge between distantly located loci [74,75], it is unsurprising that the KRAB-ZNF heterochromatin-like domains (blocks) make far *cis*- contacts that emerge as the B4 subcompartment in Hi-C maps [140]. Importantly, treating chromatin fibres as BCPs has been used with considerable success to accurately simulate contact maps derived from Hi-C experiments [203,210–214].

# 4. Heterochromatin-*like* domains/complexes and Hi-C maps

Heterochromatin-*like* domains/complexes (table 1) are contiguous with the chromatin fibre that is tightly folded within the confines of the nucleus. As part of the fibre, the domains/complexes will fold and experience a myriad of *cis*- and *trans*-chromosomal contacts. Many will be transient and of low frequency. Others will occur more frequently and endure, as will be the case for contacts mediated by HP1–H3K9me2/3 interactions (figure 6*d*). A measure of folding can be assessed by Hi-C, a high-throughput technique that generates contact frequency (Hi-C) maps [215]. Hi-C maps derived from *bulk* populations of cells revealed the first folding paradigm—the well-known checkerboard (or plaid) pattern of contact enrichment [86] (figure 7*a*). The pattern is cell-type-specific [217,218] and identifies a set of loci that interact both in *cis* and *trans* between megabasepair-sized genomic intervals that can be classified on the basis of computational correlation and principal component analysis as either an A or B compartment, where there is higher contact frequency between genomic loci of the same type (A–A and B–B type contacts) and reduced contact frequency between loci of different types (A–B type contacts) [86]. Characterization showed that A-type compartments carry euchromatic marks, are gene rich, transcriptionally active and early replicating [219]. By contrast, B-type compartments were found to carry heterochromatic marks, are gene poor, late replicating and often associated with the nuclear lamina [219]. On this basis, the checkerboard contact pattern is thought to represent the folding of chromatin into euchromatin (A-type compartments) and heterochromatin (B-type compartments) [86].

Contacts between heterochromatin-*like* domains/complexes should segregate with the B-type heterochromatic compartments. Large domains do. As explained, the B4 heterochromatic subcompartment contains the KRAB–ZNF genes from the clusters on human chromosome 19 that are assembled into heterochromatin-*like* domains [138,140] (table 1). However, the checkerboard pattern in Hi-C maps [86,216] (figure 7*a*) does not show signs of the smaller domains/complexes (table 1). They may now have been observed in Hi-C maps generated from *Nipbl*$^{-/-}$ liver cells where chromatin-associated cohesin was depleted [216] (figure 7*e*). This revealed a finer heterochromatic B-type compartmentalization that *emerged* from the A-type compartments; canonical B-type compartments did not exhibit fragmentation. Notably, loci in the A-type compartments retained their euchromatic epigenetic marks, i.e. A-type loci do not turn into B-type loci in *Nipbl*$^{-/-}$ cells. The absence of the finer B-like compartments in wt Hi-C maps is because B–B contacts involving loci within the finer B-type compartments are continuously disrupted by ATP-dependent loop extrusion, whereupon the loci segregate with the A-type compartments. It is only when chromatin-associated cohesin is depleted that those contacts are re-instated and the finer, innate, B-type compartments emerge from the A-type compartments [216].

Cohesin is a loop-extruding factor (LEF) [212,214]. LEFs attach to the chromatin fibre at random positions and reel it in from both sides, thereby extruding a progressively growing chromatin loop (figure 7, top row) until they either fall off, bump into each other, or bump into extrusion barriers such as CTCF, which define TAD boundaries [212,214,216]. Loop extrusion is an energy-driven, ATP-dependent, process [220]. Based on the known activity of LEFs, a simple explanation for the 'masking' of the finer, innate, B-type compartments

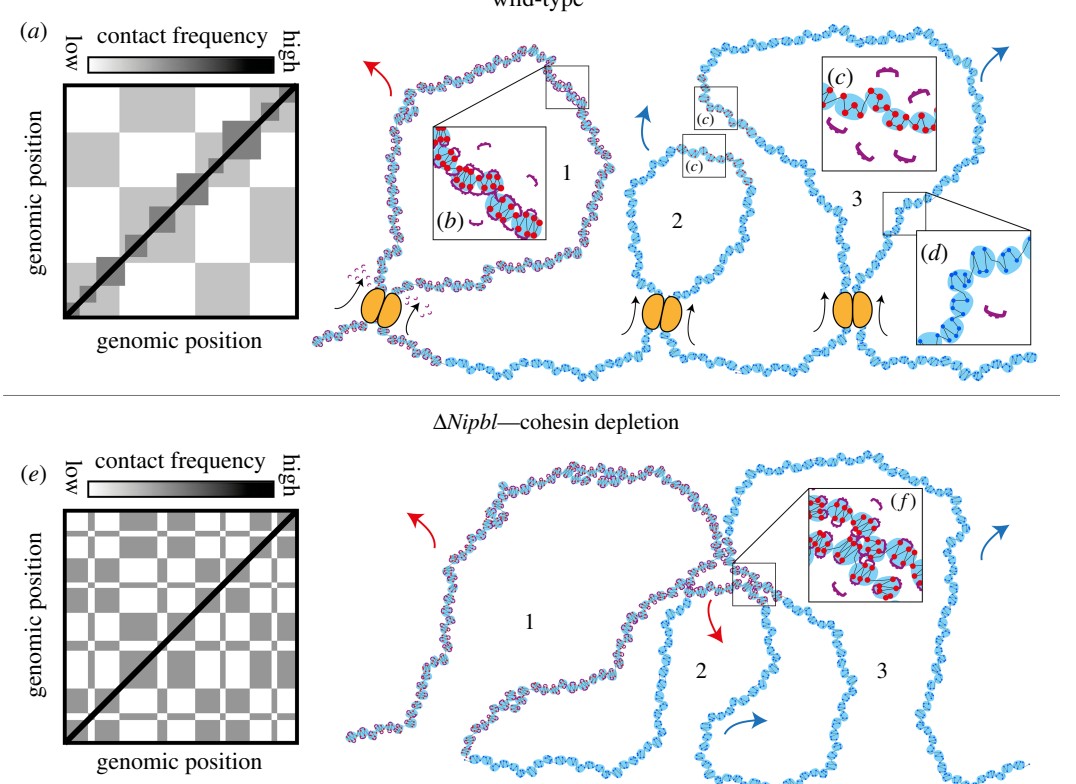

**Figure 7.** Heterochromatin-*like* domains/complexes and the emergence of finer compartmental domains in *Nipbl*$^{-/-}$ cells. In (*a*) is a cartoon representation of a wild-type (wt) Hi-C pattern of interphase chromatin organization. The strength of each pixel indicates the relative pairwise contact probability of two loci. Along the diagonal are squares of increased contact frequency that represent TADs. The off-diagonal checkerboard pattern represents compartmentalization. To the right of (*a*) is a mechanistic model of how contacts between heterochromatin-*like* domains/complexes contribute to the compartmentalization in wt cells (depicted in (*a*)). There are three loops (labelled 1, 2 and 3) that form owing to LEF-driven (ATP-dependent) loop extrusion (LEFs are given as yellow ovals; extrusion given by black arrows). Loop 1 is a homo-polymer entirely composed of 'clutches' of H3K9me2/3-marked nucleosomes (given as red circles) that are organized as two-start helix, where the second-nearest nucleosomes are preferentially bridged by HP1 proteins; there is also bridging between clutches (depicted in *b*). As loop 1 is extruded by the LEFs, there is some disruption of the HP1-mediated bridging of H3K9me2/3-marked nucleosomes within and between the 'clutches' (as depicted by the unbound HP1 dimers surrounding the LEFs), but the size and homogeneity of loop 1 makes the HP1 bridging largely resistant to disruption by ATP-driven loop extrusion. Loop 1 will make contacts with (far) *cis*- and *trans*-loci (red arrow) that will be seen as an increase in contact probability in B-type heterochromatic compartments. Loop 1 is connected to loop 2 *via* 'clutches' of 'euchromatic' nucleosomes (given as blue circles) that exhibit less of the two-start contact geometry of H3K9me2/3-marked nucleosomes and possess a more disordered or heterogeneous organization (depicted in (*d*)). Loops 2 and 3 contain smaller heterochromatin-*like* complexes that behave like small blocks in a block co-polymer. Owing to LEF-driven (ATP-dependent) loop extrusion the HP1-mediated bridging of H3K9me2/3-marked nucleosomes within and between the 'clutches' of the smaller complexes are disrupted (cf. (*c*) with (*b*)). The H3K9me2/3-marked 'clutches' take on the character of the 'euchromatic' clutches, where the constituent nucleosomes are more disorganised and not bound by HP1, as seen in (*c*). In polymer physics theory, the block co-polymer (loops 2 and 3) will have said to have undergone mixing and become 'homogeneous' with respect to euchromatin. As a consequence, both loops 2 and 3 will be dominated by far *cis*- and *trans*-contacts with A-type loci (blue arrows), which will be seen as an increase in contact probability in A-type compartments; loci within the small heterochromatin-like complexes will now fall into A-type compartments. In (*e*) is a cartoon representation of the Hi-C pattern after depletion of the LEF cohesin in *Nipbl*$^{-/-}$ cells [216]. The TADs disappear and a fine-scale compartmentalization emerges that is more defined compared to the wt situation (cf. (*e*) with (*a*)). To the right of (*e*) is a mechanistic model that depicts how contacts between heterochromatin-*like* domains/complexes contribute to the finer compartmentalization in *Nipbl*$^{-/-}$ cells (depicted in (*e*)). The same three loops (labelled 1, 2 and 3) as in the top row, but LEF-driven (ATP-dependent) loop extrusion is absent. This has little effect on the organization of loop 1, which is a homo-polymer made of 'clutches' containing H3K9me2/3 marked nucleosomes that are bridged by HP1 (depicted in (*b*)). The effect of cohesin depletion on the smaller heterochromatin-*like* complexes in loops 2 and 3 is the key to understanding the finer compartmentalization observed. In the absence of cohesin, the energy-driven disruption of HP1-mediated bridging of H3K9me2/3-marked nucleosomes is stopped and the smaller

heterochromatin-like complexes are reconstituted (cf. (*f*) with (*c*)). The newly reconstituted heterochromatin-*like* complexes de-mix and phase separate as observed for BCPs. In (*f*), *cis*-interactions between the larger heterochromatin-like domain (loop 1) and the two newly reconstituted complexes ('blocks'; from loops 2 and 3) are shown. These contacts will be detected in Hi-C experiments as the finer B-type compartments that emerge from the A-type compartments. The red arrows represent potential far *cis*- and *trans*-interactions of HP1-bridged H3K9me3-marked nucleosomes. The blue arrows represent potential far *cis*- and *trans*-interaction of 'euchromatic' nucleosomes. Contact frequency cartoons in (*a*) and (*e*) are taken and modified from [212].

in wt cells and their emergence from A-type compartments in cohesin-depleted cells can be posited drawing upon polymer physics, where heterochromatin-*like* domains/complexes are treated as one of the blocks in a BCP [209] (figure 6). The general principles can be illustrated by analytical treatment of a bulk BCP made up of two blocks A and B, where electrostatic interactions are negligible. Micro-phase separation is dependent on three parameters: (i) the volume fraction of the blocks A and B ($f_A + f_B = 1$), (ii) the total degree of polymerization ($N = N_A + N_B$), and (iii) the Flory–Huggins parameter, $\chi_{AB}$ [221]. The $\chi_{AB}$-parameter is key, because it specifies the degree of incompatibility between the A and B blocks and this is what ultimately drives micro-phase separation; a positive $\chi_{AB}$-parameter means the blocks are incompatible and the larger $\chi_{AB}$ is, the more incompatible they are. The degree of micro-phase separation of the BCP is determined by the segregation product, $\chi_{AB}N$. Given that the incompatibility is significant (i.e. the energy cost of mixing A and B is high; $\chi_{AB}$ is positive) and $\chi_{AB}N > 10.5$, the BCP will de-mix and micro-phase separate. Because the $\chi$-parameter varies inversely with thermal energy, an increase in temperature decreases the incompatibility between the constituent blocks and mixing takes place; de-mixing and micro-phase separation resumes upon cooling. In a similar way, LEF-dependent (ATP-consuming) extrusion of a chromatin loop containing a heterochromatin-*like* domain/complex (block) embedded within euchromatin results in mixing of the domain/complex with euchromatin. Mixing takes place because the energy-driven extrusion disrupts the HP1-mediated 'bridging' of H3K9me3-marked nucleosomes (figure 7*c*); incompatibility is reduced as the 'block' becomes less 'heterochromatic' and more 'euchromatic'. Moreover, because phase separation of BCPs is dependent upon volume fraction, mixing has a greater effect on smaller complexes than larger domains (cf. figure 7*b* with *c*). Put simply, mixing of smaller heterochromatin-like complexes with larger regions of surrounding euchromatin during loop extrusion will effectively make loops 'homogeneous' with respect to euchromatin. This will promote incorporation of smaller complexes into A-type compartments (see blue arrows in the top row of figure 7). When energy-driven loop extrusion is inhibited by cohesin depletion (analogous to cooling a BCP), HP1 'bridging' of H3K9me2/3-marked nucleosomes within the complexes is reconstituted resulting in de-mixing and micro-phase separation (figure 7*f*). Even the smallest heterochromatin-*like* complexes are likely to phase separate; polymer simulations and chromatin fragmentation experiments indicate that the minimal size of a chromatin 'block' required for phase separation is around 6–20 kb [211,222]. The newly micro-phase-separated complexes (blocks) can then engage in *cis*- and *trans*- contacts mediated by HP1 'bridging' of H3K9me2/3-marked nucleosomes (figure 7*f* and red arrows in the bottom row of figure 7). It is these contacts that emerge from the A-type compartments as the finer B-type heterochromatic compartments [216]. If heterochromatin-*like* domain/complexes throughout the genome are likewise subject to mixing by LEF activity, compartmentalization detected by Hi-C experiments would be coarser and less well defined in wt cells. This is indeed what is observed when wt and cohesin-depleted Hi-C maps are compared (cf. figure 7*a* with *e*) [216].

# 5. Conclusion and perspectives

HP1-mediated bridging of H3K9me2/3-marked nucleosomes provides a mechanism that connects epigenetics, PPPS and compartmentalization. The bridging of H3K9me2/3-marked nucleosomes by HP1 is involved in nucleation and assembly of heterochromatin-*like* domains/complexes that epigenetically regulate chromatin-templated processes (table 1 and figures 2–5). Bridging promotes PPPS, where macro-phase separation takes place in constitutive heterochromatin and micro-phase separation with heterochromatin-*like* domains/complexes (figures 5*a* and 6). Contacts that result from HP1-mediated bridging of H3K9me2/3-marked nucleosomes are probably detected in Hi-C maps as loci that fall within B-type compartmental domains (figure 7). Based on these data, testable predictions and explanations can be posited that could provide insight into chromatin-templated processes and genome compartmentalization. We also redefine B-type heterochromatic compartmental domains as *epigenetic compartmental domains* (ECDs) that represent the 'epigenetic' component of cellular identity.

## 5.1. Heterochromatin protein 1-mediated bridging of H3K9me2/3 nucleosome fibres and compartmentalization

Bridging of H3K9me2/3 nucleosome fibres by HP1 proteins is likely to contribute to compartmentalization detected in Hi-C maps (figure 7) and to that observed with cytologically visible constitutive heterochromatin (figure 6). To test the role of HP1 proteins in compartmentalization, it would be necessary to delete all three HP1 proteins in mammalian cells. HP1α/β/γ null ES die around a week after deletion of the genes (J. Sharif, A.G. Newman, P.B. Singh 2019, unpublished result), but conditional deletion of all three HP1 isotypes has been achieved in bi-potential mouse embryonic liver (BMEL) cells [223]. BMEL cells represent a system to test the role of heterochromatin-*like* domains/complexes on compartmentalization seen in Hi-C maps (is there (partial) collapse of B-type compartments in HP1α/β/γ null BMEL cells?) and whether domains/complexes affect LEF-dependent loop extrusion dynamics.

HP1α/β/γ null BMEL cells can also be used to investigate cytologically visible compartmentalization, specifically the functional relevance of constitutive heterochromatin positioning at the nuclear periphery. In conventional nuclei, constitutive heterochromatin is found at three locations in the interphase nucleus—internally as large domains called chromocenters, adjacent to nucleoli or at the nuclear periphery [224]. When peripheral heterochromatin is experimentally untethered, it re-localizes from the periphery to the nuclear interior and coalesces with other domains to form a single large phase-separated block of heterochromatin [225,226]. Strikingly, despite the obvious change in nuclear organization, compartmentalization as detected in Hi-C experiments is unchanged [226]. This raised the question of what the functional relevance of peripheral heterochromatin is. It was recently suggested that tethering of constitutive heterochromatin to the periphery inhibits aggregation of macro-phase-separated constitutive heterochromatin domains into the single large phase-separated block that is observed in the 'untethered' nuclei [225,226]. The factors that are likely to promote aggregation include molecular crowding [2,4] and bridging molecules such as HP1 proteins (figures 2 and 3). The tendency to aggregate can be stopped by tethering constitutive heterochromatin to a larger structure, such as the nuclear lamina. Heterochromatin anchored to the periphery could further resist aggregation of internal heterochromatin domains through fibres that emanate from the periphery to the internal domains. A prediction of this simple model is that the chromatin fibres connecting the periphery to the internal domains are 'spring-loaded' and deletion of HP1 proteins, which removes one of the forces driving aggregation, would lead to movement of the internal domains towards the periphery as the fibres relax. This is, in fact, what is observed in HP1α/β/γ null BMEL cells (fig. 3D in [223]). HP1α/β/γ null BMEL nuclei also show a reduction in H3K9me3, albeit many nuclei still possess dense constitutively heterochromatic domains, indicating that other interactions, in addition to the HP1–H3K9me3 interaction, are involved in macro-phase separation of constitutive heterochromatin. As explained, these could include, *inter alia*, other bridging molecules [3] or an affinity between homotypic DNA repetitive elements [165].

There are examples in nature where aggregation of heterochromatin into a single mass is programmed as part of normal development. In nocturnal mammals, rod photoreceptors have non-conventional nuclei where heterochromatin aggregates into a single internally located block, which acts as a micro-lens to facilitate nocturnal vision [227]. Here, aggregation is an advantage; the process of *maturation* or *hardening* of the phase-separated block could, moreover, lead to the formation of a more solid glassy/liquid-crystalline phase [62] that might possess lens-like properties. Such being the case, natural selection will have seen to it that peripheral heterochromatin became untethered from the nuclear lamina in rod photoreceptors. This is what appears to have happened [225,227].

## 5.2. Heterochromatin-*like* domains/complexes and self-assembly of block copolymers

Micro-phase separation of BCPs in solution can result in self-assembly into a wide variety of nanostructures [228], including vesicles, rods and liquid-crystalline lamellae [229]. The ability of heterochromatin-*like* domains/complexes to likewise adopt different micro-phases will influence their properties and function. For example, heterochromatin-*like* domains/complexes that behave as liquid crystals [230] provide an explanation for heterochromatic PEV, where a heterochromatic gene variegates when placed in a euchromatic environment [231]. A liquid-crystalline micro-phase could be promoted by the atypical organization of heterochromatic genes, where their introns are replete with middle repetitive sequences similar to those located in regions of peri-centric heterochromatin [232]. Viewing domains/complexes as liquid crystals might also provide an explanation for the proximity effects [230] seen with variegating heterochromatic genes [233], and the sole example of dominant PEV, *brown-dominant* ($bw^D$) variegation [234].

## 5.3. Heterochromatin-like domains/complexes and the phylotypic stage of vertebrate development

The phylotypic stage of vertebrate development represents the archetype of the basic body plan, where there is high morphological similarity between different vertebrate species [235–237]. This stage represents the 'bottle-neck' in the hour-glass model of development, where embryos exhibit greater variation at the earliest stages and at later stages but at the phylotypic stage, there is an evolutionary restriction (bottle-neck) in which only a reduced amount of evolution is allowed and, as a consequence, the morphologies of the embryos are similar [238]. It is unclear how this mid-embryogenesis stage of development has come to be the most conserved, but it is thought to involve the activity of conserved transcription/chromatin factors and developmental signalling pathways where perturbations in these mechanisms have fatal consequences, thus leading to evolutionary conservation [239,240]. Recent work has shown that H3K9me3-marked heterochromatin is deployed transiently in germ layer cells—overlapping with the phylotypic stage—to repress genes associated with fully differentiated cell function and then, as development proceeds, to undergo reorganization and loss during lineage specification [144,241]. Assembly of differentiation-specific genes into H3K9me3-marked heterochromatin may represent one of the conserved chromatin-based mechanisms that regulate the 'phylotypic restriction'. It is tempting to speculate that the H3K9me3-marked heterochromatin assembled in germ layer cells may contain heterochromatin-*like* domains/complexes (table 1) and like the domains/complexes is subject to HP1 'bridging' and PPPS (figure 6).

PPPS of H3K9me3-marked heterochromatin could explain its resistance to sonication after chemical cross-linking [143]. Sonication-resistant heterochromatin (srHC) has been isolated from human fibroblasts and sequenced [143,241] and shown to contain the B4 subcompartment (figure 5*b,c*). We have found that the 1.2 Mb superTAD, which contains greater than 70 genes at the clustered Protocadherin (cPcdh) locus (table 1) [142] is also found in srHC [143]. The cPcdh exons are expressed combinatorially in neurons and the Protocadherin proteins that arise from this process form multimers that interact homophilically and mediate a variety of developmental processes, including neuronal survival, synaptic maintenance, axonal tiling and dendritic self-avoidance [242]. The Protocadherin superTAD is regulated by the KRAB-ZFP pathway and, notably, ablation of the SETDB1 H3K9HMTase leads to collapse of the entire superTAD [142]. It will be of interest to investigate whether srHC isolated from germ layer cells contain KRAB-ZFP-regulated TADs and compartments. This would indicate a role for KRAB-ZFPs and the heterochromatin-*like* domain/complexes they assemble in regulating the 'phylotypic restriction' during vertebrate development.

## 5.4. 'Epigenetic compartmental domains' and the regulation of cellular identity

At the outset, it was posited that the HP1-containing heterochromatin-*like* domains/complexes would be mechanistically related to Polycomb (Pc)-containing domains/complexes [169,243,244]. The similarity between HP1- and *Pc*-domains/complexes extends to their ability to phase separate and to generate B-type heterochromatic compartmental domains in Hi-C maps. For one, the mammalian *Pc* homologue CBX2, like HP1 proteins, can promote phase separation that is dependent upon amino acids in CBX2 necessary for nucleosome fibre compaction [245,246]. Second, the histone modifications H3K9me2/3 and H3K27me3 are diagnostic for HP1- and *Pc*-dependent domains/complexes, respectively (table 1) [247] and are used, *inter alia*, to define B-type compartmental domains [219]. Given that H3K9me2/3 and H3K27me3 are the only histone modifications that are truly epigenetic [247], we suggest that the B-type compartmental domains generated by HP1- and Pc-dependent domains/complexes are *epigenetic compartmental domains* (ECDs) that drive the compartmentalization seen in Hi-C maps in the same way that cytologically visible compartmental segregation is driven *solely* by heterochromatin [226].

ECDs could have functional significance. ECDs may represent the 'epigenetic' component of cellular identity where the other component, tissue-specific gene expression, is represented by contacts that generate the euchromatic A-type compartments. In this way, compartmentalization observed in Hi-C maps decouples epigenetics from tissue-specific gene expression. There is evidence that supports a role for HP1 proteins in regulating the 'epigenetic' component of cellular identity. It comes from studies on mammalian HP1β, which mediates contacts between H3K9me2/3-marked nucleosomes (figure 2, [31]) and thus likely to contribute to ECDs. HP1β is necessary for maintaining pluripotency of ES cells and the differentiated state of fibroblasts [248]. The cellular identities of two different cell types expressing divergent patterns of gene expression are both safeguarded by HP1β. In the absence of HP1β, cellular identity becomes unstable. It would seem that perturbation of contacts between heterochromatin-*like* domains/complexes might increase the cellular plasticity of differentiated cells, which should in turn enhance their reprogrammability by iPS reprogramming factors. In this regard,

RNAi screens for genes whose inhibition enhance reprogramming efficiency identified genes encoding CAF-1, the SUMO-conjugating enzyme UBE2i, SUMO2, SETDB1, ATRX and DAXX proteins [172,173]. Strikingly, all are involved in either nucleation or replication of heterochromatin-*like* domains/ complexes (table 1 and figures 1*g* and 4*a*; [95]). A prediction would be that RNAi-inhibition of these genes would perturb ECDs in Hi-C experiments. This remains to be tested.

## Availability of data and materials

Bioinformatic methods: Coordinates for genes within the B4 subcompartment annotation was obtained using the UCSC table browser, which was then reduced to unique positions using simple shell commands. Reads from ChIPseq were trimmed for quality using Trimmomatic SE and aligned to GRCh37/hg19 using Bowtie2 [249]. KAP1 peaks were called using MACS. Read coverage was normalized to reads per genomic context (1×), input-subtracted and plotted using deeptools [250]. The entire pipeline was deployed using Snakemake [251]. Genomic coverage was visualized using the IGV genome browser. The reprogramming-resistant regions (RRRs) annotation was graciously provided by Dr Yi Zhang, which was converted to human coordinates using the UCSC's liftover tool. Hi-C contact frequency map for chromosome 19 in H1-ESCs was obtained by using the 3D genome browser [252].

| Data | publication | GEO acc. |
| --- | --- | --- |
| input subtracted H3K9me3 ChIP from euchromatin and sonication-resistant heterochromatin | Becker *et al.* [143] PMID: 29272703 | GSE87041 |
| HP1β and control ChIP in HEK293 | LeRoy *et al.* [253] PMID: 22897906 | GSE39579 |
| H3K9me3 and KAP1 ChIP and input in naive ES cells | Theunissen *et al.* [254] PMID: 27424783 | GSE84382 |
| B4 subcompartment annotation (modified to include chr19:19 775 198–24 317 418—Vogel *et al.* [138]) | Rao *et al.* [140] PMID:25497547 | GSE63525 |
| Hi-C contact map in H1-ESCs | Dixon *et al.* [217] PMID:25693564 | GSE52457 |

Data accessibility. See the Availability of data and materials section.
Authors' contributions. P.B.S. conceived of the synthesis presented and wrote the first draft. A.G.N. undertook the bio-informatic analyses and drew the figures. Both authors revised and approved the final manuscript.
Competing interests. The authors declare that they have no competing interests.
Funding. This work was supported by a grant from the Ministry of Education and Science of Russian Federation (grant no. 14.Y26.31.0024); P.B.S. was also supported by Nazarbayev University Grant 090118FD5311. A.G.N. was supported by Deutsche Forschungsgemeinschaft (DFG)—Projekt number 410579311.
Acknowledgements. We acknowledge the BIH HPC cluster for use of their computing infrastructure.

## Glossary

| | |
| --- | --- |
| ADD | ATRX-DNMT3-DNMT3 L domain |
| ATRX | Alpha Thalassemia/Mental Retardation Syndrome X-Linked |
| BCP | block copolymer |
| BMEL | bi-potential mouse embryonic liver |
| CAF-1 | chromatin assembly factor 1 |
| *Cbx1, 3, 5* | *Chromobox* homologue *1, 3* and *5* encoding HP1β, γ and α proteins, respectively |
| CD | chromodomain |
| CSD | chromo shadow domain |
| cPcdh | clustered Protocadherin |
| CTCF | CCCTC-binding factor |
| DamID | DNA adenine methyltransferase identification |
| DAXX | Death Domain-associated protein |
| DNMTases | DNA methyltransferases |
| DNMT1 | maintenance DNA methyltransferase 1 |
| DNMT3A | de novo DNA methyltransferase 3A |
| DNMT3B | de novo DNA methyltransferase 3B |

| | |
|---|---|
| DNMT3 L | DNA methyltransferase 3 L |
| ECD | epigenetic compartmental domain |
| ES | embryonic stem |
| FISH | fluorescent *in situ* hybridization |
| FRAP | fluorescent recovery after photobleaching |
| G9a | G9a K9H3 HMTase |
| GLP | G9a-like K9H3 HMTase |
| gDMRs | germline differentially methylated regions |
| H3K9me2/3 | *di/tri*-methylated lysine 9 on histone H3 |
| H3K27me3 | *tri*-methylated lysine 27 on histone H3 |
| H4K20me3 | *tri*-methylated lysine 20 on histone H4 |
| HMTases | histone methyltransferases |
| HDACs | histone deacetylases |
| HR | hinge region |
| HP1 | heterochromatin protein 1 |
| iPS | induced pluripotent stem cell |
| *I(s)* | interaction probability |
| KAP1 | KRAB-associated protein 1 |
| kb | kilobases |
| $k_B$ | Boltzmann's constant |
| KRAB-ZNF | KRAB domain-zinc finger |
| KRAB-ZFP | KRAB domain-zinc finger protein |
| KRIP1 | KRAB-A interacting protein-1 |
| *lacO* | lac operator |
| lacI protein (encoded by *lacR* gene) | inhibitor of the lactose operon |
| LLPS | liquid–liquid phase separation |
| LEFs | loop extrusion factors |
| Mb | megabases |
| Np95 | nuclear protein 95 |
| NuRD | nucleosome remodelling histone deacetylase |
| PEV | position-effect variegation |
| PHD | plant homeodomain |
| pLI | loss of function intolerance |
| Pc | polycomb |
| PPPS | polymer–polymer phase separation |
| PxVxL | proline/any/valine/any/leucine pentapeptide motif |
| RBCC | ring-finger B box-coiled coil domain |
| RRR | reprogramming-resistant regions |
| SCNT | somatic cell nuclear transfer |
| SETDB1 | SET domain bifurcated 1 K9H3 HMTase |
| siRNA | small-interfering RNA |
| SMARCAD1 | SWI/SNF-related, matrix-associated actin-dependent regulator of chromatin, subfamily A, containing DEAD/H box 1 |
| srHC | sonication-resistant heterochromatin |
| SUMO2 | small ubiquitin-related modifier 2 |
| SUV39H1/2 | mammalian Suvar K9H3 HMTase 1 and 2 |
| *T* | temperature |
| $T_H1$ | type 1 T-helper cell |
| $T_H2$ | type 2 T-helper cell |
| TAD | topologically associated domain |
| Tif1β | transcriptional intermediary factor 1-β |
| TRIM28 | tripartite motif-containing 28 |
| UBE2i | ubiquitin conjugating enzyme 2i |
| wt | wild-type |
| $\chi$ | the Flory–Huggins parameter |
| Zscan4 | zinc finger and SCAN domain-containing 4 |

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
