## [Reviewer comments · Royal Society Open Science]

Review History

RSOS-191976.R0 (Original submission)

Review form: Reviewer 1

Is the manuscript scientifically sound in its present form?

Yes

Are the interpretations and conclusions justified by the results?

Yes

Is the language acceptable?

Yes

Do you have any ethical concerns with this paper?

No

Have you any concerns about statistical analyses in this paper?

No

Recommendation?

Major revision is needed (please make suggestions in comments)

Comments to the Author(s)

Singh and Newman provide a review of the state of the knowledge regarding the 3D conformation of chromosomes and epigenetics. I think this is a difficult task and they have done a commendable job. They do not provide new data/code/method/modelling and it appears that the only original contribution is to rename compartments in "Epigenetic compartmental domains". Because of this, I will referee this paper as if it was a review.

1. ".. could favour the merging of "clutches" to form ordered (fractal) "globules" that have been detected .. " is incorrect. Entropy works towards the equilibrium globule state. There is no direct connection between crowding/depletion attraction and fractal globule which is instead a non-equilibrium state of a polymer.

2. The authors do a very good job at connecting experiments with modelling work in many instances. They also correctly point out that LLPS is different from PPPS and that the latter is most likely the mechanism of phase separation in the nucleus.

I have a few more suggestions

a. Regarding HP1 bridges:

Barbieri et al PNAS 2013

Brackley et al PNAS 2012

are to my knowledge the first using explicit protein bridges to study the folding of chromosomes.

b. An important work connecting co-polymer models with epigenetics is that of Daniel Jost et al NAR 2014. It was the first to my knowledge to make this connection clear.

c. Regarding the spreading of epigenetic marks driven by (HP1) bridges + writers see Dodd et al "Theoretical analysis of epigenetic cell memory by nucleosome modification." Cell 2007

Michieletto et al "Polymer model with Epigenetic Recoloring Reveals a Pathway for the de novo Establishment and 3D Organization of Chromatin Domains" PRX 2016

Adachi and Kawaguchi "Chromatin state switching in a polymer model with mark-conformation coupling" PRE 2019

d. Recent works extending these spreading models to non-equilibrium polymer-polymer micro-phase separation are

Michieletto et al "Nonequilibrium Theory of Epigenomic Microphase Separation in the Cell Nucleus" PRL 2019

Coli et al "Magnetic polymer models for epigenetics-driven chromosome folding" PRE 2019

4. Cross-linking typically has an irreversible connotation so I would not use it to describe the reversible bridging of HP1 or other proteins on the chromatin.

5. In several instances, it should be made clear that this is a review and that no new data is presented but simply reported from previous papers or obtained from online tools such as the genome browser.

6. Do the authors have permission to reuse Figs.7A-E and other panels that are taken from published work?

Review form: Reviewer 2

Is the manuscript scientifically sound in its present form?

Yes

Are the interpretations and conclusions justified by the results?

Yes

Is the language acceptable?

Yes

Do you have any ethical concerns with this paper?

No

Have you any concerns about statistical analyses in this paper?

No

Recommendation?

Accept with minor revision (please list in comments)

Comments to the Author(s)

Singh and Newman (MS ID#RSOS-191976)

In this review article, the authors describe how heterochromatin protein 1 (HP1) controls/orchestrates heterochromatin formation or heterochromatin compartmentalization. Especially, they propose that polymer-polymer phase separation (PPPS) is the major driving force of heterochromatin compartmentalization induced by HP1. Also, their interpretation of the Hi-C A/B compartment data in cohesion depleted cells is interesting and quite insightful. This article is definitely general interest to wide range of biologists and valuable to publish in ROYAL SOCIETY OPEN SCIENCE. The reviewer hopes the authors should address following comments before publication,

1. PPPS vs LLPS

In page 6, line 6-11

The authors said that “if HP1a does drive phase separation it does so in a manner different from multivalent intrinsically-disordered proteins [63], which are known to form endogenous liquid-liquid phase separated condensates in the nucleus [70]. Based on these data we propose an alternative mechanism. Mammalian HP1 proteins drive phase separation by polymer-polymer phase separation (PPPS; [71]) rather than by LLPS.”

The reviewer thinks this interpretation is too biased. Reference [70] showed that only HP1a forms liquid droplets in vitro, but this does not deny the possibility of other HP1 isoforms as a critical component of phase-separated heterochromatin compartment induced by LLPS. Indeed, reference [69] showed such possibility. Also, LLPS-mediated but different compartments exist, suggesting that different intrinsically-disordered proteins which contribute to LLPS not always form same droplet/compartment. Data of ref [70] also does not deny that HP1-mediated heterochromatin may exist as LLPS or formation of LLPS may contribute to function or maintenance of heterochromatin. Therefore, the PPPS story is fine and interesting, but the authors should more carefully describe current situation and cite recent work of HP1 function in LLPS appropriately. Also, most recent Narlikar’s nature paper (Sanulli et al 2019) should be integrated into this article.

2. Fig. 7

Page 14 first paragraph. The authors should cite more details of each panel for description of Fig. 7 to help understanding the authors points.

Also, if any paper shows that HP1 level on H3K9me2/3+ high regions in compartment A is lower than that in compartment B, should be cited because this support the authors’ Fig7 model.

3. conclusions and perspectives

It is nice to add one additional fig to show some of issues discussed in this section and how heterochromatin-like domains contribute to biological functions and evolution.

3. minor

Page 4, line 51-54.

"Cb_{x5}^{-/-} mice are viable and fertile (cited in [48]) [47] albeit they exhibit a very specific defect where TH1 expression is not silenced in TH2 cells [49]."

"TH1 expression" should be "TH1 specific gene expression" or "TH1 function".

Decision letter (RSOS-191976.R0)

08-Jan-2020

Dear Dr Singh,

The editors assigned to your paper ("On the relation of phase separation and Hi-C maps to epigenetics") have now received comments from reviewers.

Both reviewers are very positive about publication of your review, but each of them has some substantive suggestions for improving the paper before we can proceed to consider accepting the manuscript. We would like you to revise your paper in accordance with the referee suggestions which can be found below (not including confidential reports to the Editor). Please note this decision does not guarantee eventual acceptance.

Please submit a copy of your revised paper before 31-Jan-2020. Please note that the revision deadline will expire at 00.00am on this date. If we do not hear from you within this time then it will be assumed that the paper has been withdrawn. In exceptional circumstances, extensions may be possible if agreed with the Editorial Office in advance. We do not allow multiple rounds of revision so we urge you to make every effort to fully address all of the comments at this stage. If deemed necessary by the Editors, your manuscript will be sent back to one or more of the original reviewers for assessment. If the original reviewers are not available, we may invite new reviewers.

If your study uses humans or animals please include details of the ethical approval received, including the name of the committee that granted approval. For human studies please also detail

whether informed consent was obtained. For field studies on animals please include details of all permissions, licences and/or approvals granted to carry out the fieldwork.

- Data accessibility

If you wish to submit your supporting data or code to Dryad (<http://datadryad.org/>), or modify your current submission to dryad, please use the following link:
<http://datadryad.org/submit?journalID=RSOS&manu=RSOS-191976>

- Competing interests

- Authors' contributions

- Acknowledgements

- Funding statement

Kind regards,

Andrew Dunn

on behalf of Prof Steve Brown (Subject Editor)
 openscience@royalsociety.org

Comments to Author:

Reviewers' Comments to Author:

Reviewer: 1

Comments to the Author(s)

Singh and Newman provide a review of the state of the knowledge regarding the 3D conformation of chromosomes and epigenetics. I think this is a difficult task and they have done a commendable job. They do not provide new data/code/method/modelling and it appears that the only original contribution is to rename compartments in "Epigenetic compartmental domains". Because of this, I will referee this paper as if it was a review.

1. "... could favour the merging of "clutches" to form ordered (fractal) "globules" that have been detected .. " is incorrect. Entropy works towards the equilibrium globule state. There is no direct connection between crowding/depletion attraction and fractal globule which is instead a non-equilibrium state of a polymer.

2. The authors do a very good job at connecting experiments with modelling work in many instances. They also correctly point out that LLPS is different from PPPS and that the latter is most likely the mechanism of phase separation in the nucleus.

I have a few more suggestions

a. Regarding HP1 bridges:

Barbieri et al PNAS 2013

Brackley et al PNAS 2012

are to my knowledge the first using explicit protein bridges to study the folding of chromosomes.

b. An important work connecting co-polymer models with epigenetics is that of Daniel Jost et al NAR 2014. It was the first to my knowledge to make this connection clear.

c. Regarding the spreading of epigenetic marks driven by (HP1) bridges + writers see Dodd et al "Theoretical analysis of epigenetic cell memory by nucleosome modification." Cell 2007

Michieletto et al "Polymer model with Epigenetic Recoloring Reveals a Pathway for the de novo Establishment and 3D Organization of Chromatin Domains" PRX 2016

Adachi and Kawaguchi "Chromatin state switching in a polymer model with mark-conformation coupling" PRE 2019

d. Recent works extending these spreading models to non-equilibrium polymer-polymer micro-phase separation are

Michieletto et al "Nonequilibrium Theory of Epigenomic Microphase Separation in the Cell Nucleus" PRL 2019

Coli et al "Magnetic polymer models for epigenetics-driven chromosome folding" PRE 2019

4. Cross-linking typically has an irreversible connotation so I would not use it to describe the reversible bridging of HP1 or other proteins on the chromatin.

5. In several instances, it should be made clear that this is a review and that no new data is presented but simply reported from previous papers or obtained from online tools such as the genome browser.

6. Do the authors have permission to reuse Figs.7A-E and other panels that are taken from published work?

Reviewer: 2

Comments to the Author(s)

Singh and Newman (MS ID#RSOS-191976)

In this review article, the authors describe how heterochromatin protein 1 (HP1) controls/orchestrates heterochromatin formation or heterochromatin compartmentalization. Especially, they propose that polymer-polymer phase separation (PPPS) is the major driving force of heterochromatin compartmentalization induced by HP1. Also, their interpretation of the Hi-C A/B compartment data in cohesion depleted cells is interesting and quite insightful. This article is definitely general interest to wide range of biologists and valuable to publish in ROYAL SOCIETY OPEN SCIENCE. The reviewer hopes the authors should address following comments before publication,

1. PPPS vs LLPS

In page 6, line 6-11

The authors said that “if HP1a does drive phase separation it does so in a manner different from multivalent intrinsically-disordered proteins [63], which are known to form endogenous liquid-liquid phase separated condensates in the nucleus [70]. Based on these data we propose an alternative mechanism. Mammalian HP1 proteins drive phase separation by polymer-polymer phase separation (PPPS; [71]) rather than by LLPS.”

The reviewer thinks this interpretation is too biased. Reference [70] showed that only HP1a forms liquid droplets in vitro, but this does not deny the possibility of other HP1 isoforms as a critical component of phase-separated heterochromatin compartment induced by LLPS. Indeed, reference [69] showed such possibility. Also, LLPS-mediated but different compartments exist, suggesting that different intrinsically-disordered proteins which contribute to LLPS not always form same droplet/compartment. Data of ref [70] also does not deny that HP1-mediated heterochromatin may exist as LLPS or formation of LLPS may contribute to function or maintenance of heterochromatin. Therefore, the PPPS story is fine and interesting, but the authors should more carefully describe current situation and cite recent work of HP1 function in LLPS appropriately. Also, most recent Narlikar’s nature paper (Sanulli et al 2019) should be integrated into this article.

2. Fig. 7

Page 14 first paragraph. The authors should cite more details of each panel for description of Fig. 7 to help understanding the authors points.

Also, if any paper shows that HP1 level on H3K9me2/3+ high regions in compartment A is lower than that in compartment B, should be cited because this support the authors’ Fig7 model.

3. conclusions and perspectives

It is nice to add one additional fig to show some of issues discussed in this section and how heterochromatin-like domains contribute to biological functions and evolution.

3. minor

Page 4, line 51-54.

“Cbx5-/- mice are viable and fertile (cited in [48]) [47] albeit they exhibit a very specific defect where TH1 expression is not silenced in TH2 cells [49].”

“TH1 expression” should be “TH1 specific gene expression” or “TH1 function”.

Author's Response to Decision Letter for (RSOS-191976.R0)

See Appendix A.

RSOS-191976.R1 (Revision)

Review form: Reviewer 1

Is the manuscript scientifically sound in its present form?

Yes

Are the interpretations and conclusions justified by the results?

Yes

Is the language acceptable?

Yes

Do you have any ethical concerns with this paper?

No

Have you any concerns about statistical analyses in this paper?

No

Recommendation?

Accept with minor revision (please list in comments)

Comments to the Author(s)

The authors have made an effort to address some of my comments. Still, I feel that this is a strange review in that it is not limited to provide an overview of current work in the field but also appears to propose new models. Some of them are not new although they are presented as such using misleading wording, e.g. ".. in our scheme .." pg 12 line 4, "we propose an alternative mechanism .." pg 6 ln 9, "we suggest that tethering .." pg 17 ln 35, "we suggest that the B-type compartmental ..." pg 19 ln 50, etc).

I feel that I would not do my job as a reviewer if I didn't point out that the authors should be clear and careful in attributing merit.

I'd warmly encourage the authors to reword ambiguous sentences, such as,

- we suggest that tethering -> It was recently suggested that tethering ...
- we propose an alternative mechanism -> A an alternative mechanism was proposed
- in our scheme -> remove
- we suggest that the B-type compartmental .. -> Existing models suggest that B-compartments may be associated to epigenetic marks such as HP1 and Pc (and add references to, e.g., Rao Cell 2014, di Pierro PNAS 2017, Michieletto PRL 2019, ..., where the idea of the relationship between compartments and epigenetics is actually being tested experimentally and with computational models).

Review form: Reviewer 2

Is the manuscript scientifically sound in its present form?

No

Are the interpretations and conclusions justified by the results?

No

Is the language acceptable?

Yes

Do you have any ethical concerns with this paper?

No

Have you any concerns about statistical analyses in this paper?

No

Recommendation?

Accept as is

Comments to the Author(s)

Unfortunately the author basically ignore the comment 1 which is only the critical issue from the reviewer. " the authors should more carefully describe current situation and cite recent work of HP1 function in LLPS appropriately." No further comment.

Decision letter (RSOS-191976.R1)

27-Jan-2020

Dear Dr Singh:

On behalf of the Editors, I am pleased to inform you that your Manuscript RSOS-191976.R1 entitled "On the relations of phase separation and Hi-C maps to epigenetics" has been accepted for publication in Royal Society Open Science subject to minor revision in accordance with the referee suggestions. Please find the referees' comments at the end of this email.

The reviewers have recommended publication, but also suggest some minor revisions to your manuscript. I invite you to respond to the comments and revise your manuscript. It is important to note that both reviewers raise some critical comments that will require further consideration and revision, for example one of the reviewer's comments about precision on attribution. Please deal with each of these comments and ensure that you have properly considered each of the requests of the reviewers.

- Ethics statement

- Data accessibility

<http://datadryad.org/submit?journalID=RSOS&manu=RSOS-191976.R1>

- Competing interests

- Authors' contributions

- Acknowledgements

- Funding statement

Because the schedule for publication is very tight, it is a condition of publication that you submit the revised version of your manuscript before 05-Feb-2020. Please note that the revision deadline will expire at 00.00am on this date. If you do not think you will be able to meet this date please let me know immediately.

To revise your manuscript, log into <https://mc.manuscriptcentral.com/rsos> and enter your Author Centre, where you will find your manuscript title listed under "Manuscripts with Decisions". Under "Actions," click on "Create a Revision." You will be unable to make your

revisions on the originally submitted version of the manuscript. Instead, revise your manuscript and upload a new version through your Author Centre.

on behalf of Prof Steve Brown (Subject Editor)
openscience@royalsociety.org

Reviewer comments to Author:
Reviewer: 2

Comments to the Author(s)
Unfortunately the author basically ignore the comment 1 which is only the critical issue from the reviewer. " the authors should more carefully describe current situation and cite recent work of HP1 function in LLPS appropriately."
No further comment.

Reviewer: 1

Comments to the Author(s)

The authors have made an effort to address some of my comments. Still, I feel that this is a strange review in that it is not limited to provide an overview of current work in the field but also appears to propose new models. Some of them are not new although they are presented as such using misleading wording, e.g. ".. in our scheme .." pg 12 line 4, "we propose an alternative mechanism .." pg 6 ln 9, "we suggest that tethering .." pg 17 ln 35, "we suggest that the B-type compartmental ..." pg 19 ln 50, etc).

I feel that I would not do my job as a reviewer if I didn't point out that the authors should be clear and careful in attributing merit.

I'd warmly encourage the authors to reword ambiguous sentences, such as,

- we suggest that tethering -> It was recently suggested that tethering ...
- we propose an alternative mechanism -> A an alternative mechanism was proposed
- in our scheme -> remove
- we suggest that the B-type compartmental .. -> Existing models suggest that B-compartments may be associated to epigenetic marks such as HP1 and Pc (and add references to, e.g., Rao Cell 2014, di Pierro PNAS 2017, Michieletto PRL 2019, ..., where the idea of the relationship between compartments and epigenetics is actually being tested experimentally and with computational models).

Author's Response to Decision Letter for (RSOS-191976.R1)

See Appendix B.

Decision letter (RSOS-191976.R2)

03-Feb-2020

Dear Dr Singh,

It is a pleasure to accept your manuscript entitled "On the relations of phase separation and Hi-C maps to epigenetics" in its current form for publication in Royal Society Open Science. There were no further comments from reviewers or editors on your manuscript.

on behalf of Steve Brown (Subject Editor)
openscience@royalsociety.org

Appendix A

Dr Andrew Dunn,
Royal Society Open Science Editorial Office,
Royal Society Open Science,
openscience@royalsociety.org

Re: Manuscript ID RSOS-191976

Dear Andrew,

Please find below our point-by-point responses to the reviewers (our replies are in red). The changes made to the uploaded revised manuscript are also given in red.

Reviewers' Comments to Author:

We thank the reviewers for their thoughtful comments.

Reviewer: 1

Comments to the Author(s)

Singh and Newman provide a review of the state of the knowledge regarding the 3D conformation of chromosomes and epigenetics. I think this is a difficult task and they have done a commendable job. They do not provide new data/code/method/modelling and it appears that the only original contribution is to rename compartments in "Epigenetic compartmental domains". Because of this, I will referee this paper as if it was a review.

1. "... could favour the merging of "clutches" to form ordered (fractal) "globules" that have been detected .. " is incorrect. Entropy works towards the equilibrium globule state. There is no direct connection between crowding/depletion attraction and fractal globule which is instead a non-equilibrium state of a polymer.

Accepted. We removed any connection between crowding/depletion attraction and ordered fractal globules and instead spoken of crowding/depletion attraction, dense domains and globules. We have added new references to make this clear. The sentence now reads:

"....."clutches" to form "dense domains" or "globules" that have been detected by super-resolution imaging (Nozki et al., 2017; [8]) and chromosome conformation capture (Rao et al., 2017; [9])."

2. The authors do a very good job at connecting experiments with modelling work in many instances. They also correctly point out that LLPS is different from PPPS and that the latter is most likely the mechanism of phase separation in the nucleus.

Accepted.

I have a few more suggestions

a. Regarding HP1 bridges:

Barbieri et al PNAS 2013

Brackley et al PNAS 2012

are to my knowledge the first using explicit protein bridges to study the folding of chromosomes.

b. An important work connecting co-polymer models with epigenetics is that of Daniel Jost et al NAR 2014. It was the first to my knowledge to make this connection clear.

c. Regarding the spreading of epigenetic marks driven by (HP1) bridges + writers see
Dodd et al "Theoretical analysis of epigenetic cell memory by nucleosome modification." Cell 2007
Micheletto et al "Polymer model with Epigenetic Recoloring Reveals a Pathway for the de novo Establishment and 3D Organization of Chromatin Domains" PRX 2016
Adachi and Kawaguchi "Chromatin state switching in a polymer model with mark-conformation coupling" PRE 2019

d. Recent works extending these spreading models to non-equilibrium polymer-polymer micro-phase separation are

Micheletto et al "Nonequilibrium Theory of Epigenomic Microphase Separation in the Cell Nucleus" PRL 2019

Coli et al "Magnetic polymer models for epigenetics-driven chromosome folding" PRE 2019

We are grateful to the reviewer for suggesting these models and are respectful of his/her scholarship. For what it's worth, some models regarding HP1 and Pc proteins were proposed by ourselves during an earlier epoch (Singh and Huskisson, Developmental Genetics, 1998). Given that we already have ~250 references we focussed on those references suggested by reviewer 1 that are key. Accordingly, we have added Barbieri et al PNAS 2012 and Daniel Jost et al NAR 2014 to the references at references 87 and 165 respectively.

4. Cross-linking typically has an irreversible connotation so I would not use it to describe the reversible bridging of HP1 or other proteins on the chromatin.

Accepted. We have used "bridging" instead of "cross-linking" where appropriate in the text.

5. In several instances, it should be made clear that this is a review and that no new data is presented but simply reported from previous papers or obtained from online tools such as the genome browser.

With all due respects to the reviewer, this is an odd request. Yes, the paper is submitted as a review; the cover to the manuscript, generated by the journal, states under "Article type" that it is.

That said, in keeping with the guidelines to authors, it is more than a simple review: it is a synthesis that has explanatory force and makes specific testable predictions.

The synthesis presented is founded upon mammalian HP1 proteins and H3K9me2/3-marked heterochromatin-like domains/complexes they assemble (see Table 1), which places current knowledge on phase separation and Hi-C maps in mammalian cells within a single paradigm that can be tested experimentally. It has explanatory force and posits testable predictions for a variety of chromatin-templated phenomena ranging from cytologically-visible compartmentalisation, through epigenetic compartmental domains and the mechanisms that maintain cellular identity to the evolution of phylotypic restriction during mammalian development.

Our synthesis enabled us, for the first time, to make a prediction as to the contacts that likely result in macro- and micro-phase separation by PPPS. We have also confirmed that there are likely to be enough nucleation sites within the B4 sub-compartment for it to be inherited as an epigenetic domain.

Experiment will determine the veracity of this paradigm.

6. Do the authors have permission to reuse Figs.7A-E and other panels that are taken from published work?

Yes.

Panel A of Figure 1 is taken and modified from Hiragami-Hamada, et al., (2016) and is used according to the Creative Commons License CC BY that allows for maximum dissemination and re-use of open access materials. The source is properly acknowledged.

Panels B through to G of Figure 1 are from Billur et al., (2010). PBS is corresponding author on that paper and re-use was granted by Elsevier through Copyright Clearance Center's RightsLink service. Figure 2 is taken and modified from Machida et al., (2018) and re-use was granted by Elsevier through Copyright Clearance Center's RightsLink service.

Panels in A and E of Figure 7 are taken from Neubler et al., (2018) and are used according to the Creative Commons license CC BY-NC-ND that enables re-use of figures with appropriate acknowledgement of the original article and mention of any changes made; the work is funded by the US government and is in the public domain by default.

Reviewer: 2

Comments to the Author(s)

Singh and Newman (MS ID#RSOS-191976)

In this review article, the authors describe how heterochromatin protein 1 (HP1) controls/orchestrates heterochromatin formation or heterochromatin compartmentalization. Especially, they propose that polymer-polymer phase separation (PPPS) is the major driving force of heterochromatin compartmentalization induced by HP1. Also, their interpretation of the Hi-C A/B compartment data in cohesion depleted cells is interesting and quite insightful. This article is definitely general interest to wide range of biologists and valuable to publish in ROYAL SOCIETY OPEN SCIENCE. The reviewer hopes the authors should address following comments before publication,

Accepted. We would add that it is only because we propose that HP1 proteins regulate PPPS that it is possible to gain insight into the changes observed in “*the Hi-C A/B compartments data in cohesion (sic) depleted cells*”.

1. PPPS vs LLPS

In page 6, line 6-11

The authors said that “if HP1a does drive phase separation it does so in a manner different from multivalent intrinsically-disordered proteins [63], which are known to form endogenous liquid-liquid phase separated condensates in the nucleus [70]. Based on these data we propose an alternative mechanism. Mammalian HP1 proteins drive phase separation by polymer-polymer phase separation (PPPS; [71]) rather than by LLPS.”

The reviewer thinks this interpretation is too biased. Reference [70] showed that only HP1a forms liquid droplets in vitro, but this does not deny the possibility of other HP1 isoforms as a critical component of phase-separated heterochromatin compartment induced by LLPS. Indeed, reference [69] showed such possibility. Also, LLPS-mediated but different compartments exist, suggesting that different intrinsically-disordered proteins which contribute to LLPS not always form same droplet/compartment. Data of ref [70] also does not deny that HP1-mediated heterochromatin may exist as LLPS or formation of LLPS may contribute to function or maintenance of heterochromatin. Therefore, the PPPS story is fine and interesting, but the authors should more carefully describe current situation and cite recent work of HP1 function in LLPS appropriately. Also, most recent Narlikar’s nature paper (Sanulli et al 2019) should be integrated into this article.

With all due respects to reviewer 2, reviewer 1 agrees with us (see point 2 of reviewer 1 above). Also, as mentioned above, it is only because we propose that HP1 proteins mediate PPPS that it is possible to gain insight into the changes observed in “*the Hi-C A/B compartments data in cohesion (sic) depleted cells*”.

Nevertheless, there is some merit to reviewer 2’s comments and in order to assuage his/her concerns we have added the following sentences: “These data indicate that phase separation of mammalian constitutive heterochromatin is unlikely to be mediated by HP1-driven LLPS, albeit work in *Drosophila* (Strom et al., 2017; [70]) and fission yeast (Sanulli et al., 2019; [71]) shows that HP1 proteins can drive LLPS. Rather, we suggest that the major mechanism by which mammalian HP1 proteins drive phase separation is polymer-polymer phase separation (PPPS; [72]) as opposed to LLPS.”

2. Fig. 7

Page 14 first paragraph. The authors should cite more details of each panel for description of Fig. 7 to help understanding the authors points.

Also, if any paper shows that HP1 level on H3K9me2/3+ high regions in compartment A is lower than that in compartment B, should be cited because this support the authors’ Fig7 model.

We are grateful for the reviewer’s comment. We have provided very detailed legends to Figure 7 (over a page long). Also, the paragraph that goes over the page, from page 14 to 15, describes the panels in detail; we have tried to make this accessible by limiting the mathematical treatment and

giving clear and simple explanations of the panels. At the risk of repeating ourselves and taking away from the clarity we not added more to the first paragraph.

We agree that one test of our model would be to measure HP1 proteins on H3K9me2/3-marked nucleosomes in A and B compartments in wt cells and to compare this to HP1 at the same H3K9me2/3-marked nucleosomes after cohesin depletion. This experiment is beyond the scope of the current review.

3. conclusions and perspectives

It is nice to add one additional fig to show some of issues discussed in this section and how heterochromatin-like domains contribute to biological functions and evolution.

With all due respects to this reviewer, we appreciate the opportunity to add another figure but feel that the paper is already too long.

3. minor

Page 4, line 51-54.

“Cbx5^{-/-} mice are viable and fertile (cited in [48]) [47] albeit they exhibit a very specific defect where TH1 expression is not silenced in TH2 cells [49].”

“TH1 expression” should be “TH1 specific gene expression” or “TH1 function”.

Accepted. “TH1 expression” has been changed to “TH1-specific gene expression”.

Appendix B

Dr Andrew Dunn,
Royal Society Open Science Editorial Office,
Royal Society Open Science,
openscience@royalsociety.org

Re: Manuscript ID RSOS-191976R1

Dear Andrew,

Please find below our point-by-point responses to the reviewers' comments on the revised manuscript RSOS-191976_R1 (our replies are in red). We have uploaded the revised manuscript 2 (RSOS-191976_R2). In the interests of clarity RSOS-191976_R2 also shows, in red, the changes that were made in RSOS-191976_R1.

Reviewers' Comments to Author:

Reviewer: 2

Unfortunately the author basically ignore the comment 1 which is only the critical issue from the reviewer. " the authors should more carefully describe current situation and cite recent work of HP1 function in LLPS appropriately." No further comment.

We have undertaken what reviewer 2 asked of us. Specifically:

1. We have included two extra references on *Drosophila* and fission yeast HP1 homologues in relation to LLPS.
2. Regarding mammalian HP1 proteins the reviewer had stated: "*Data of ref [70] also does not deny that HP1-mediated heterochromatin may exist as LLPS or formation of LLPS may contribute to function or maintenance of heterochromatin.*" The absence of evidence does not constitute evidence.
3. Reviewer 1 agreed with us saying: "*They [the authors] also correctly point out that LLPS is different from PPPS and that the latter is most likely the mechanism of phase separation in the nucleus.*"
4. We have also softened the phraseology saying that PPPS is the major mechanism rather than the sole mechanism: "*Rather, we suggest that the major mechanism by which mammalian HP1 proteins drive phase separation is polymer-polymer phase separation (PPPS; [72]) as opposed to LLPS.*"

Reviewer: 1

Comments to the Author(s)

The authors have made an effort to address some of my comments. Still, I feel that this is a strange review in that it is not limited to provide an overview of current work in the field but also appears to propose new models. Some of them are not new although they are presented as such using misleading wording, e.g. ".. in our scheme .." pg 12 line 4, "we propose an alternative mechanism .." pg 6 ln 9, "we suggest that tethering .." pg 17 ln 35, "we suggest that the B-type compartmental ..." pg 19 ln 50, etc).

I feel that I would not do my job as a reviewer if I didn't point out that the authors should be clear

and careful in attributing merit. I'd warmly encourage the authors to reword ambiguous sentences, such as,

- we suggest that tethering -> It was recently suggested that tethering ...
- we propose an alternative mechanism -> An alternative mechanism was proposed
- in our scheme -> remove
- we suggest that the B-type compartmental .. -> Existing models suggest that B-compartments may be associated to epigenetic marks such as HP1 and Pc (and add references to, e.g., Rao Cell 2014, di Pierro PNAS 2017, Michieletto PRL 2019, ..., where the idea of the relationship between compartments and epigenetics is actually being tested experimentally and with computational models).

We are struck by the sentimentality of this reviewer's comments. Our replies are as follows:

1. We have deleted "in our scheme", albeit the scheme presented in Figure 4B, first panel, is modified from our previous work in reference [102] as stated in the figure legend.
2. With all due respects to this reviewer he/she has not read the changes we made in the first revision (RSOS-191976_R1) in response to the comments made by reviewer 2. The statement "we propose an alternative mechanism...." is no longer in the manuscript. Specifically, the sentence in the original submission: "Based on these data we propose an alternative mechanism. Mammalian HP1 proteins drive phase separation by polymer-polymer phase separation (PPPS; [71]) rather than by LLPS.", was changed in RSOS-191976_R1 to: "These data indicate that phase separation of mammalian constitutive heterochromatin is unlikely to be mediated by HP1-driven LLPS, albeit work in *Drosophila* (Strom et al., 2017; [70]) and fission yeast (Sanulli et al., 2019; [71]) shows that HP1 proteins can drive LLPS. Rather, we suggest that the major mechanism by which mammalian HP1 proteins drive phase separation is polymer-polymer phase separation (PPPS; [72]) as opposed to LLPS."
3. We have replaced "we suggest that tethering" with "It was recently suggested that tethering..."
4. We have not made this change because we are the first to define B-type compartmental domains as *epigenetic compartmental domains* that drive compartmentalisation as observed in Hi-C maps. Notably, this reviewer began his previous comments with the exhortation: "They do not provide new data/code/method/modelling and it appears that the only original contribution is to rename compartments in "Epigenetic compartmental domains". He/she now wishes to place our contribution (as he/she sees it) at the feet of others; his/her feelings may have got the better of him/her. We also note that Rao et al., 2014 is cited 6 times in the text. We have added one reference cited by this reviewer after the sentence "Importantly, treating chromatin fibres as BCPs has been used with considerable success to accurately simulate contact maps derived from Hi-C experiments [156,165–169]", where Di Perro et al., 2017 replaces Di Perro et al., 2018 [168]. We have already alluded to related work from the Marenduzzo laboratory (Michieletto et al., PRL 2019 given by the reviewer) in reference [156] (Michieletto et al., NAR 2017).